# Implicit Manifold Gaussian Process Regression

**Bernardo Fichera**[1]    **Viacheslav Borovitskiy**[2]    **Andreas Krause**[2]    **Aude Billard**[1]

[1] EPFL    [2] ETH Zürich

## Abstract

Gaussian process regression is widely used because of its ability to provide well-calibrated uncertainty estimates and handle small or sparse datasets. However, it struggles with high-dimensional data. One possible way to scale this technique to higher dimensions is to leverage the implicit low-dimensional *manifold* upon which the data actually lies, as postulated by the *manifold hypothesis*. Prior work ordinarily requires the manifold structure to be explicitly provided though, i.e. given by a mesh or be known to be one of the well-known manifolds like the sphere. In contrast, in this paper we propose a Gaussian process regression technique capable of inferring implicit structure directly from data (labeled and unlabeled) in a fully differentiable way. For the resulting model, we discuss its convergence to the Matérn Gaussian process on the assumed manifold. Our technique scales up to hundreds of thousands of data points, and improves the predictive performance and calibration of the standard Gaussian process regression in some high-dimensional settings.

## 1   Introduction

Gaussian processes are among the most adopted models for learning unknown functions within the Bayesian framework. Their data efficiency and aptitude for uncertainty quantification make them appealing for modeling and decision-making applications in the fields like robotics (Deisenroth and Rasmussen, 2011), geostatistics (Chilès and Delfiner, 2012), numerics (Hennig et al., 2015), etc.

The most widely used Gaussian process models, like squared exponential and Matérn (Rasmussen and Williams, 2006), impose the simple assumption of differentiability of the unknown function while also respecting the geometry of $\mathbb{R}^d$ by virtue of being stationary or isotropic. Such simple assumptions make uncertainty estimates *reliable*, albeit too conservative at times. The same simplicity makes these models struggle from the *curse of dimensionality*. We hypothesize that it is still possible to leverage these simple priors for real world high-dimensional problems granted that they are adapted to the implicit *low-dimensional submanifolds* where the data actually lies, as illustrated by Figure 1.

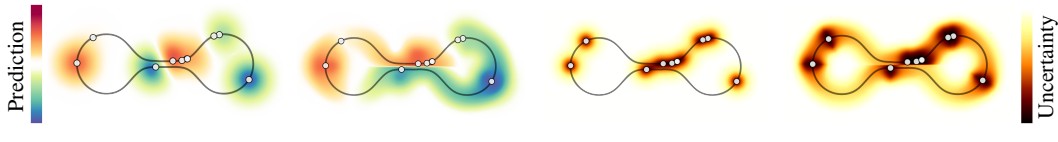

(a) Euclidean, prediction    (b) Ours, prediction    (c) Euclidean, uncertainty    (d) Ours, uncertainty

Figure 1: *Euclidean* (standard Matérn-5/2 kernel) vs *ours* (implicit manifold) Gaussian process regression for data that lies on a dumbbell-shaped curve (1-dimensional manifold) assumed unknown. The data contains a small set of labeled points and a large set of unlabeled points. Our technique recognizes that the two lines in the middle are intrinsically far away from each other, giving a much better model on and near the manifold. Far away from the manifold it reverts to the Euclidean model.

Code available at https://github.com/nash169/manifold-gp.

37th Conference on Neural Information Processing Systems (NeurIPS 2023).

Recent works in machine learning generalized Matérn Gaussian processes for modeling functions on non-Euclidean domains such as manifolds or graphs (Azangulov et al., 2022; 2023; Borovitskiy et al., 2021; 2023; 2020). Crucially, this line of work assumes *known* geometry (e.g. a manifold or a graph) beforehand. In this work we aim to widen the applicability of Gaussian processes for higher dimensional problems by *automatically learning* the implicit low-dimensional manifold upon which the data lies, the existence of which is suggested by the *manifold hypothesis*. We propose a new model which learns this structure and approximates the Matérn kernel on the implicit manifold.

Our approach can operate in both supervised and semi-supervised settings, with the emphasis on the latter: uncovering the implicit manifold may require a lot of samples from it, however these samples need not be labeled, and unlabeled data is usually more abundant. Taking inspiration in the manifold learning results of Coifman and Lafon (2006), Dunson et al. (2021) and others we approximate the unknown manifold by an appropriately weighted nearest neighbor graph. Then we use graph Matérn kernels thereon as approximations to the manifold Matérn kernels of Borovitskiy et al. (2020), extending them to the vicinity of the manifold in the ambient $\mathbb{R}^d$ in an appropriate way.

## 1.1 Related Work and Contribution

High-dimensional Gaussian process regression is an area of active research, primarily motivated by decision making applications like Bayesian optimization. There are three main directions in this area: (1) selecting a small subset of input dimensions, (2) learning a small number of new features by linearly projecting the inputs and (3) learning non-linear features. Our technique belongs to the third direction. Further details on the area can be found in the recent review by Binois and Wycoff (2022).

A close relative of our technique in the literature is described in Dunson et al. (2022). It targets the low-dimensional setting where the inputs are densely sampled on the underlying surface. It is based on the heat (diffusion) kernels on graphs as in Kondor and Lafferty (2002) and uses the Nyström method to extend kernels to $\mathbb{R}^d$, both of which may incur a high computational cost. Another close relative is the concurrent work by Peach et al. (2023) on modeling vector fields on implicit manifolds.

We are targeting the high-dimensional setting. Here, larger datasets of partly labeled points are often needed to *infer* geometry. Because of this, we emphasize computational efficiency by leveraging sparse precision matrix structure of Matérn kernels (as opposed to the heat kernels) and use KNN for sparsifying the graph and accelerating the Nyström method. This results in linear computational complexity with respect to the number of data points. Furthermore, the model we propose is fully differentiable, which may be used to find both kernel and geometry hyperparameters by maximizing the marginal likelihood. Finally, to get reasonable predictions on the whole ambient space $\mathbb{R}^d$, we combine the prediction of the geometric model with the prediction of a classical Euclidean Gaussian process, weighting these by the relative distance to the manifold.

The geometric model is differentiable with respect to its kernel-, likelihood- and geometry-related hyperparameters, with gradient evaluation cost being linear with respect to the number of data points. After training, we can efficiently compute the predictive mean and kernel as well as sample the predictive model, providing the basic computational primitives needed for the downstream applications like Bayesian optimization. We evaluate our technique on a synthetic low-dimensional example and test it in a high-dimensional large dataset setting of predicting rotation angles of rotated MNIST images, improving over the standard Gaussian process regression.

## 2 Gaussian Processes

A Gaussian process $f \sim \mathrm{GP}(m, k)$ is a distribution over functions on a set $\mathcal{X}$. It is determined by the mean function $m(\boldsymbol{x}) = \mathbb{E} f(\boldsymbol{x})$ and the covariance function (kernel) $k(\boldsymbol{x}, \boldsymbol{x}') = \mathrm{Cov}(f(\boldsymbol{x}), f(\boldsymbol{x}'))$.

Given data $\mathbf{X}, \boldsymbol{y}$, where $\mathbf{X} = (\boldsymbol{x}_1, .., \boldsymbol{x}_n)^\top$ and $\boldsymbol{y} = (y_1, .., y_n)^\top$ with $\boldsymbol{x}_i \in \mathcal{X}^d, y_i \in \mathbb{R}$, one usually assumes $y_i = f(\boldsymbol{x}_i) + \varepsilon_i$ where $\varepsilon_i \sim \mathrm{N}(0, \sigma_\varepsilon^2)$ is IID noise and $f \sim \mathrm{GP}(0, k)$ is some *prior* Gaussian process, whose mean is assumed to be zero in order to simplify notation. The posterior distribution $f \mid \boldsymbol{y}$ is then another Gaussian process $f \mid \boldsymbol{y} \sim \mathrm{GP}(\hat{m}, \hat{k})$ with (Rasmussen and Williams, 2006)

$$\hat{m}(\cdot) = \mathbf{K}_{(\cdot)\mathbf{X}}\big(\mathbf{K}_{\mathbf{X}\mathbf{X}} + \sigma_\varepsilon^2 \mathbf{I}\big)^{-1} \boldsymbol{y}, \qquad \hat{k}(\cdot, \cdot') = \mathbf{K}_{(\cdot, \cdot')} - \mathbf{K}_{(\cdot)\mathbf{X}}\big(\mathbf{K}_{\mathbf{X}\mathbf{X}} + \sigma_\varepsilon^2 \mathbf{I}\big)^{-1} \mathbf{K}_{\mathbf{X}(\cdot')}, \quad (1)$$

where the matrix $\mathbf{K}_{\mathbf{X}\mathbf{X}}$ has entries $k(\boldsymbol{x}_i, \boldsymbol{x}_j)$, the vector $\mathbf{K}_{\mathbf{X}(\cdot)} = \mathbf{K}_{(\cdot)\mathbf{X}}^\top$ has components $k(\boldsymbol{x}_i, \cdot)$. If needed, one can efficiently sample $f \mid \boldsymbol{y}$ using *pathwise conditioning* (Wilson et al., 2020; 2021)

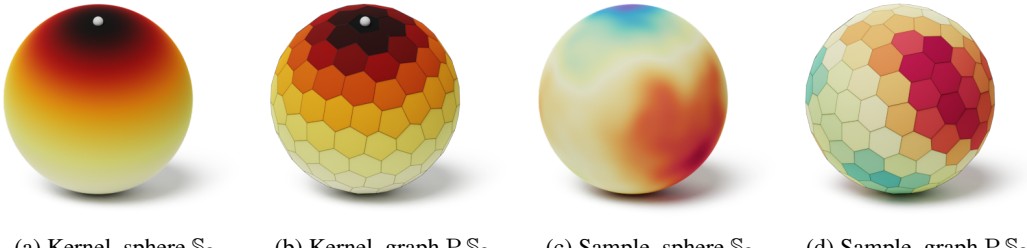

(a) Kernel, sphere $\mathbb{S}_2$     (b) Kernel, graph $\mathrm{P}\,\mathbb{S}_2$     (c) Sample, sphere $\mathbb{S}_2$     (d) Sample, graph $\mathrm{P}\,\mathbb{S}_2$

Figure 2: Kernel values $k(\bullet, \cdot)$ and samples for the Matérn-$3/2$ Gaussian processes on the sphere manifold $\mathbb{S}_2$ and for the approximating Matérn-$5/2$ process on a geodesic polyhedron graph $\mathrm{P}\,\mathbb{S}_2$.

Matérn Gaussian processes—including the limiting $\nu \to \infty$ case, squared exponential Gaussian processes—are the most popular family of models for $\mathcal{X} = \mathbb{R}^d$. These have zero mean and kernels

$$k_{\nu,\kappa,\sigma^2}(\boldsymbol{x}, \boldsymbol{x}') = \sigma^2 \frac{2^{1-\nu}}{\Gamma(\nu)} \left( \sqrt{2\nu} \frac{\|\boldsymbol{x} - \boldsymbol{x}'\|}{\kappa} \right)^\nu K_\nu \left( \sqrt{2\nu} \frac{\|\boldsymbol{x} - \boldsymbol{x}'\|}{\kappa} \right) \tag{2}$$

where $K_\nu$ is the modified Bessel function of the second kind (Gradshteyn and Ryzhik, 2014) and $\nu, \kappa, \sigma^2$ are the hyperparameters responsible for smoothness, length scale and variance, respectively. We proceed to describe how Matérn processes can be generalized to inputs $\boldsymbol{x}$ lying on *explicitly given* manifolds or graphs instead of the Euclidean space $\mathbb{R}^d$.

### 2.1 Matérn Gaussian Processes on Explicit Manifolds and Graphs

For a domain which is a Riemannian manifold, an obvious and natural idea for generalizing Matérn Gaussian processes could be to substitute the Euclidean distances $\|x - x'\|$ in Equation (2) with the geodesic distance. However, this approach results in ill-defined kernels that fail to be positive semi-definite (Feragen et al., 2015; Gneiting, 2013).

Another direction for generalization is based on the stochastic partial differential equation (SPDE) characterization of Matérn processes first described by Whittle (1963): $f \sim \mathrm{GP}(0, k_{\nu,\kappa,\sigma^2})$ solves

$$\left( \frac{2\nu}{\kappa^2} - \Delta_{\mathbb{R}^d} \right)^{\frac{\nu}{2} + \frac{d}{4}} f = \mathcal{W}, \tag{3}$$

where $\Delta_{\mathbb{R}^d}$ is the standard Laplacian operator and $\mathcal{W}$ is the Gaussian white noise with variance proportional to $\sigma^2$. If taken to be the definition, this characterization can be easily extended to general Riemannian manifolds $\mathcal{X} = \mathcal{M}$ by substituting $\Delta_{\mathbb{R}^d}$ with the Laplace–Beltrami operator $\Delta_{\mathcal{M}}$, taking $d = \dim \mathcal{M}$ and substituting $\mathcal{W}$ with the appropriate generalization of the Gaussian white noise (Lindgren et al., 2011). Based on this idea, Borovitskiy et al. (2020) showed that on *compact* Riemannian manifolds, Matérn Gaussian processes are the zero-mean processes with kernels

$$k_{\nu,\kappa,\sigma^2}(x, x') = \frac{\sigma^2}{C_{\nu,\kappa}} \sum_{l=0}^{\infty} \left( \frac{2\nu}{\kappa^2} + \lambda_l \right)^{-\nu - d/2} f_l(x) f_l(x'), \tag{4}$$

where $-\lambda_l, f_l$ are eigenvalues and eigenfunctions of the Laplace–Beltrami operator and $C_{\nu,\kappa}$ is the normalizing constant ensuring that $\frac{1}{\mathcal{X}} \int_{\mathcal{X}} k_{\nu,\kappa,\sigma^2}(x, x) \, \mathrm{d}x = \sigma^2$. This, alongside with considerations from Azangulov et al. (2022) allows one to practically compute $k_{\nu,\kappa,\sigma^2}$ for many compact manifolds.

If the domain $\mathcal{X}$ is a weighted undirected graph $\mathcal{G}$, we can also use Equation (3) to define Matérn Gaussian processes on $\mathcal{G}$ (Borovitskiy et al., 2021). In this case, $\Delta_{\mathbb{R}^d}$ is substituted with the minus graph Laplacian $-\Delta_{\mathcal{G}}$ and $\mathcal{W} \sim \mathrm{N}(0, \sigma_{\mathcal{W}}^2 \mathbf{I})$ is the vector of IID Gaussians. Here, SPDE transforms into a stochastic linear system, whose solution is of the same form as Equation (4) but with a finite sum instead of the infinite series, with $d = 0$ because there is no canonical notion of dimension for graphs and with $\lambda_l, f_l$ being the eigenvalues and eigenvectors—as functions on the node set—of the matrix $\Delta_{\mathcal{G}}$. These processes are illustrated on Figure 2.

# 3 Implicit Manifolds and Gaussian Processes on Them

Consider a dataset $\mathbf{X} = (\boldsymbol{x}_1, .., \boldsymbol{x}_N)^\top$, $\boldsymbol{x}_i \in \mathbb{R}^d$ partially labeled with labels $y_1, .., y_n \in \mathbb{R}$, $n \le N$. Assume that $\boldsymbol{x}_i$ are IID randomly sampled from a compact Riemannian submanifold $\mathcal{M} \subseteq \mathbb{R}^d$. As by Section 2.1, the manifold $\mathcal{M}$ is associated to a family of Matérn Gaussian processes tailored to its geometry. We do not assume to know $\mathcal{M}$, only the fact that it exists, hence the question is: how can we recover the kernels of the aforementioned geometry-aware processes from the observed dataset?

It is clear from Equation (4) that to recover $k_{\nu,\kappa,\sigma^2}$ we need to get the eigenpairs $-\lambda_l$, $f_l$ of the Laplace–Beltrami operator on $\mathcal{M}$. Naturally, for a finite dataset this can only be done approximately. We proceed to discuss the relevant theory of Laplace–Beltrami eigenpair approximation.

## 3.1 Background on Approximating the Eigenpairs of the Laplace–Beltrami Operator

There exists a number of theoretical and empirical results on eigenpair approximation. Virtually all of them study approximating the implicit manifold by some kind of a weighted undirected graph[1] with node set $\{\boldsymbol{x}_1, .., \boldsymbol{x}_N\}$ and weights that are somehow determined by the Euclidean distances $\|\boldsymbol{x}_i - \boldsymbol{x}_j\|$. The eigenvalues of the *graph Laplacian* on this graph are supposed to approximate the eigenvalues of the Laplace–Beltrami operator, while the eigenvectors—regarded as functions on the node set—approximate the values of the eigenfunctions of the Laplace–Beltrami operator at $\boldsymbol{x}_i \in \mathcal{M}$. To approximate eigenfunctions elsewhere, any sort of continuous (smooth) interpolation suffices.

There are three popular notions of graph Laplacian. Let us denote the adjacency matrix of the weighted graph by $\mathbf{A}$ and define $\mathbf{D}$ to be the diagonal degree matrix with $\mathbf{D}_{ii} = \sum_j \mathbf{A}_{ij}$. Then

$$\underbrace{\boldsymbol{\Delta}_{\text{un}} = \mathbf{D} - \mathbf{A}}_{\text{unnormalized}}, \qquad \underbrace{\boldsymbol{\Delta}_{\text{sym}} = \mathbf{I} - \mathbf{D}^{-1/2}\mathbf{A}\mathbf{D}^{-1/2}}_{\text{symmetric normalized}}, \qquad \underbrace{\boldsymbol{\Delta}_{\text{rw}} = \mathbf{I} - \mathbf{D}^{-1}\mathbf{A}}_{\text{random walk normalized}}. \qquad (5)$$

The first two of these are symmetric positive semi-definite matrices, the third is, generally speaking, non-symmetric. However, from the point of view of linear operators, all of them can be considered symmetric (self-adjoint) positive semi-definite: the first two with respect to the standard Euclidean inner product $\langle \cdot, \cdot \rangle$, and the third one with respect to the modified inner product $\langle \boldsymbol{v}, \boldsymbol{u} \rangle_{\mathbf{D}} = \langle \mathbf{D}\boldsymbol{v}, \boldsymbol{u} \rangle$. Thus for each there exists an orthonormal basis of eigenvectors and eigenvalues are non-negative.[2]

The most common way to define the graph is by setting $\mathbf{A}_{ij} = \exp\left(-\|\boldsymbol{x}_i - \boldsymbol{x}_j\|^2/4\alpha^2\right)$ for an $\alpha > 0$. If $\boldsymbol{x}_i$ are IID samples from the *uniform* distribution on the manifold $\mathcal{M}$, then all of the graph Laplacians, each multiplied by an appropriate power of $\alpha$, converge to the Laplace–Beltrami operator, both pointwise (Hein et al., 2007) and *spectrally* (García Trillos et al., 2020), i.e. in the sense of eigenpair convergence, at least at the node set of the graph.[3] However, if the inputs $\boldsymbol{x}_i$ are sampled non-uniformly, graph Laplacians, at best, converge to different continuous limits, none of which coincides with the Laplace–Beltrami operator (Hein et al., 2007).

Coifman and Lafon (2006) proposed a clever trick to handle non-uniformly sampled data $\boldsymbol{x}_1, .., \boldsymbol{x}_N$. Starting with $\tilde{\mathbf{A}}$ and $\tilde{\mathbf{D}}$ defined in the same way as $\mathbf{A}$ and $\mathbf{D}$ before, they define $\mathbf{A} = \tilde{\mathbf{D}}^{-1}\tilde{\mathbf{A}}\tilde{\mathbf{D}}^{-1}$. Intuitively, this corresponds to normalizing by the kernel density estimator to cancel out the unknown density. The corresponding $\boldsymbol{\Delta}_{\text{rw}}$ then converges pointwise to the Laplace–Beltrami operator (Hein et al., 2007), though $\boldsymbol{\Delta}_{\text{un}}$ and $\boldsymbol{\Delta}_{\text{sym}}$ do not: they converge to different continuous limits. Dunson et al. (2021, Theorem 2) show that, under technical regularity assumptions, eigenvalues $\lambda_k$ and renormalized eigenvectors of $\boldsymbol{\Delta}_{\text{rw}}$ converge to the respective eigenvalues and eigenfunctions of the Laplace–Beltrami operator, regardless of the sampling density of $\boldsymbol{x}_i$.

Both in the simple case and in the sampling density independent case, the graphs and their respective Laplacians turn out to be dense, requiring a lot of memory to store and being inefficient to operate with. To make computations efficient, sparse graphs such as KNN graphs are much more preferable over the dense graphs. Spectral convergence for KNN graphs is studied, for example, in Calder and Trillos (2022), for $\mathbf{A}_{ij} = h(\|\boldsymbol{x}_i - \boldsymbol{x}_j\|/\alpha)$ with a compactly supported regular function $h$, and with

---

[1]Other possibilities include linear (classical PCA) or quadratic (Pavutnitskiy et al., 2022) approximations. See the books by Lee and Verleysen (2007) and Ma and Fu (2011) for additional context.

[2]Naturally, for the random walk normalized Laplacian $\Delta_{\text{rw}}$ the orthonormality is with respect to $\langle \cdot, \cdot \rangle_{\mathbf{D}}$.

[3]García Trillos et al. (2020) do not explicitly study $\boldsymbol{\Delta}_{\text{sym}}$. Since $\boldsymbol{\Delta}_{\text{sym}}$ and $\boldsymbol{\Delta}_{\text{rw}}$ are *similar* matrices, they share eigenvalues, so eigenvalue convergence for $\boldsymbol{\Delta}_{\text{sym}}$ is trivial, the eigenvector convergence, however, is not.

limit depending on the sampling density. Unfortunately, we are unaware of any spectral convergence results in the literature that hold for KNN graphs and are independent of the data sampling density.

## 3.2 Approximating Matérn Kernels on Manifolds

Here we incorporate various convergence results, including but not limited to the ones described in Section 3.1, proving that all spectral convergence results imply the convergence of graph Matérn kernels to the respective manifold Matérn kernels.

**Proposition 1.** *Denote the eigenpairs by $\lambda_l, f_l$ for a graph Laplacian and by $\lambda_l^{\mathcal{M}}, f_l^{\mathcal{M}}$ for the Laplace–Beltrami operator. Fix $\delta > 0$. Assume that, with probability at least $1 - \delta$, for all $\varepsilon > 0$, for $\alpha$ small enough and for $N$ large enough we have $|\lambda_l - \lambda_l^{\mathcal{M}}| < \varepsilon$ and $|f_l(\boldsymbol{x}_i) - f_l^{\mathcal{M}}(\boldsymbol{x}_i)| < \varepsilon$. Then, with probability at least $1 - \delta$, we have $k_{\nu,\kappa,\sigma^2}^{N,\alpha,L}(\boldsymbol{x}_i, \boldsymbol{x}_j) \to k_{\nu,\kappa,\sigma^2}(\boldsymbol{x}_i, \boldsymbol{x}_j)$ as $\alpha \to 0$, $N, L \to \infty$, where*

$$k_{\nu,\kappa,\sigma^2}^{N,\alpha,L}(\boldsymbol{x}_i, \boldsymbol{x}_j) = \frac{\sigma^2}{C_{\nu,\kappa}} \sum_{l=0}^{L-1} \left( \frac{2\nu}{\kappa^2} + \lambda_l \right)^{-\nu - \dim(\mathcal{M})/2} f_l(\boldsymbol{x}_i) f_l(\boldsymbol{x}_j). \tag{6}$$

*Proof.* First prove that the tail of the series in Equation (4) converges uniformly to zero, then combine this with eigenpair bounds. See details in Appendix A. □

**Remark.** The convergence in $\boldsymbol{x}_i \in \mathcal{M}$ can be lifted to pointwise convergence for all $\boldsymbol{x} \in \mathcal{M}$ if eigenvectors are interpolated Lipschitz-continuously, simply because the eigenfunctions are smooth.

Inspired by this theory, we proceed to present the implicit manifold Gaussian process model.

## 3.3 Implicit Manifold Gaussian Process

Guided by the theory we described in the previous section we are now ready to formulate the implicit manifold Gaussian process model. Given the dataset $\boldsymbol{x}_1, .., \boldsymbol{x}_N \in \mathbb{R}^d$ and $y_1, .., y_n \in \mathbb{R}$, we put

$$\mathbf{A} = \tilde{\mathbf{D}}^{-1} \tilde{\mathbf{A}} \tilde{\mathbf{D}}^{-1}, \quad \tilde{\mathbf{A}}_{ij} = S_K(\boldsymbol{x}_i, \boldsymbol{x}_j) \exp\left( -\frac{\|\boldsymbol{x}_i - \boldsymbol{x}_j\|^2}{4\alpha^2} \right), \quad \tilde{\mathbf{D}}_{ij} = \begin{cases} \sum_m \tilde{\mathbf{A}}_{im} & i = j, \\ 0 & i \neq j. \end{cases} \tag{7}$$

Here $S_K(\boldsymbol{x}_i, \boldsymbol{x}_j) = 1$ if $\boldsymbol{x}_i$ is one of the $K$ nearest neighbors of $\boldsymbol{x}_j$ or vice versa and $S_K(\boldsymbol{x}_i, \boldsymbol{x}_j) = 0$ otherwise; all matrices are of size $N \times N$ and depend on $\alpha$ and $K$ as hyperparameters. Thanks to the coefficient $S_K(\boldsymbol{x}_i, \boldsymbol{x}_j)$ that performs KNN sparsification, the matrix $\mathbf{A}$ is sparse when $K \ll N$.[4]

Then we consider the operator $\boldsymbol{\Delta}_{\mathrm{rw}} = \mathbf{I} - \mathbf{D}^{-1}\mathbf{A}$ defined by Equation (5), whose matrix is also sparse. Denoting its eigenvalues—ordered from the smallest to the largest—by $0 = \lambda_0 \leq \lambda_1 \leq \ldots \leq \lambda_{N-1}$, and its eigenvectors—orthonormal under the modified inner product $\langle \cdot, \cdot \rangle_D$ and regarded as functions on the node set of the graph—by $f_0, f_1, \ldots, f_{N-1}$, we define Matérn kernel on graph nodes $\boldsymbol{x}_i$ by

$$k_{\nu,\kappa,\sigma^2}^{\mathbf{X}}(\boldsymbol{x}_i, \boldsymbol{x}_j) = \frac{\sigma^2}{C_{\nu,\kappa}} \sum_{l=0}^{L-1} \Phi_{\nu,\kappa}(\lambda_l) f_l(\boldsymbol{x}_i) f_l(\boldsymbol{x}_j), \qquad \Phi_{\nu,\kappa}(\lambda) = \left( \frac{2\nu}{\kappa^2} + \lambda \right)^{-\nu}, \tag{8}$$

where $L$ does not need to be equal to the actual number $N$ of eigenpairs. Doing so means truncating the high frequency eigenvectors ($f_l$ for $l$ large), which always contribute less to the sum because they correspond to smaller values of $\Phi_{\nu,\kappa}(\lambda_l)$. This can massively reduce the computational costs.

By Proposition 1, Equation (8) approximates the manifold Matérn kernel with smoothness $\nu' = \nu - \dim(\mathcal{M})/2$. We adopt such a reparametrization because it does not require estimating the a priori unknown $\dim(\mathcal{M})$. This, however, makes the typical assumption of $\nu \in \{1/2, 3/2, 5/2\}$ inadequate. We chose a particular graph Laplacian normalization, namely the random walk normalized graph Laplacian $\boldsymbol{\Delta}_{\mathrm{rw}}$, to approximate the true Laplace-Beltrami operator regardless of the potential non-uniform sampling of $\boldsymbol{x}_1, .., \boldsymbol{x}_N$, based on the theoretical insights described in Section 3.1.

The kernel in Equation (8) is only defined on the set of nodes $\boldsymbol{x}_i$, next step is to extend it to the whole space $\mathbb{R}^d$. Extending kernels is usually a difficult problem because one has to worry about positive

---

[4]Note: $\tilde{\mathbf{A}}_{ii} = 1$, as if the graph has loops. Assuming $\tilde{\mathbf{A}}_{ii} = 0$ would lead to discontinuities at later stages.

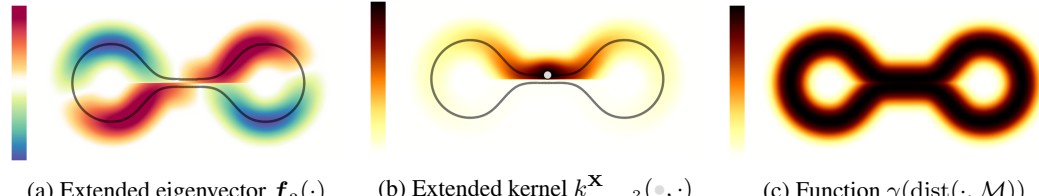

(a) Extended eigenvector $\boldsymbol{f}_3(\cdot)$     (b) Extended kernel $k^{\mathbf{X}}_{\nu,\kappa,\sigma^2}(\bullet,\cdot)$     (c) Function $\gamma(\mathrm{dist}(\cdot,\mathcal{M}))$

Figure 3: Different quantities connected to kernel extension. Notice that the values on subfigures (a) and (b) are artificially restricted to the set $\mathrm{dist}(\cdot,\mathcal{M}) < 3\alpha$ to maintain numerical stability.

semi-definiteness. To work around it, we extend the features $f_l$. For this, we use Nyström method: we allow the first argument of $S_K$ to be an arbitrary vector from $\mathbb{R}^d$, defining $S_K(\boldsymbol{x},\boldsymbol{x}_j)=1$ if $\boldsymbol{x}_j$ is one of the $K$ nearest neighbors of $\boldsymbol{x}$ among $\boldsymbol{x}_1,..,\boldsymbol{x}_N$. This allows us to extend $\tilde{\mathbf{A}}, \tilde{\mathbf{D}}, \mathbf{A}$ and $\mathbf{D}$ as

$$\tilde{A}(\boldsymbol{x},\boldsymbol{x}_j) = S_K(\boldsymbol{x},\boldsymbol{x}_j)\exp\left(-\frac{\|\boldsymbol{x}-\boldsymbol{x}_j\|^2}{4\alpha^2}\right), \quad \tilde{D}(\boldsymbol{x}) = \sum_{j=1}^{N}\tilde{A}(\boldsymbol{x},\boldsymbol{x}_j) = \sum_{\boldsymbol{x}_j\in\mathrm{KNN}(\boldsymbol{x})}\tilde{A}(\boldsymbol{x},\boldsymbol{x}_j), \quad (9)$$

$$A(\boldsymbol{x},\boldsymbol{x}_j) = \frac{\tilde{A}(\boldsymbol{x},\boldsymbol{x}_j)}{\tilde{D}(\boldsymbol{x})\tilde{D}(\boldsymbol{x}_j)}, \qquad\qquad D(\boldsymbol{x}) = \sum_{j=1}^{N}A(\boldsymbol{x},\boldsymbol{x}_j) = \sum_{\boldsymbol{x}_j\in\mathrm{KNN}(\boldsymbol{x})}A(\boldsymbol{x},\boldsymbol{x}_j), \quad (10)$$

where $\mathrm{KNN}(\boldsymbol{x})$ is the set of the $K$ nearest neighbors from $\boldsymbol{x}$ among $\boldsymbol{x}_1,..,\boldsymbol{x}_N$. With this, we define

$$f_l(\boldsymbol{x}) = \frac{1}{1-\lambda_l}\sum_{j=1}^{N}\frac{A(\boldsymbol{x},\boldsymbol{x}_j)}{D(\boldsymbol{x})}f_l(\boldsymbol{x}_j) = \frac{1}{1-\lambda_l}\sum_{\boldsymbol{x}_j\in\mathrm{KNN}(\boldsymbol{x})}\frac{A(\boldsymbol{x},\boldsymbol{x}_j)}{D(\boldsymbol{x})}f_l(\boldsymbol{x}_j). \quad (11)$$

It is easy to check that this extension perfectly reproduces the values $f_l(\boldsymbol{x}_j)$, simply because $A(\boldsymbol{x},\boldsymbol{x}_j)$ coincides with $\mathbf{A}_{ij}$ when $\boldsymbol{x}=\boldsymbol{x}_i$ and because $(f_l(\boldsymbol{x}_1),..,f_l(\boldsymbol{x}_N))^\top$ is the eigenvector of $\mathbf{A}$ corresponding to the eigenvalue $1-\lambda_l$. It also allows us to extend the kernel $k^{\mathbf{X}}_{\nu,\kappa,\sigma^2}$ as well, such that $k^{\mathbf{X}}_{\nu,\kappa,\sigma^2}(\boldsymbol{x},\boldsymbol{x}')$ is defined for arbitrary $\boldsymbol{x},\boldsymbol{x}' \in \mathbb{R}^d$ and Equation (8) still holds for the nodes $\boldsymbol{x}_1,..,\boldsymbol{x}_N$. We visualize an extended eigenvector and an extended kernel in Figures 3a and 3b.

For $\boldsymbol{x}$ far away from the nodes $\boldsymbol{x}_j$ the values of $\tilde{D}(\boldsymbol{x})$ become very small, making the extension procedure numerically unstable. Furthermore, the geometric model is generally not so relevant far away from the manifold. Because of this, the final predictive model $f^{(p)} \sim \mathrm{GP}(m^{(p)}, k^{(p)})$ combines the geometric model $f^{(m)} \sim \mathrm{GP}(m^{(m)}, k^{(m)})$—the posterior under the kernel $k^{\mathbf{X}}_{\nu,\kappa,\sigma^2}$ we defined just above—with the standard Euclidean model $f^{(e)} \sim \mathrm{GP}(m^{(e)}, k^{(e)})$—the posterior under the standard Euclidean Gaussian process in $\mathbb{R}^d$, for instance with the squared exponential kernel:

$$f^{(p)}(\boldsymbol{x}) = \gamma(\boldsymbol{x})f^{(m)}(\boldsymbol{x}) + (1-\gamma(\boldsymbol{x}))f^{(e)}(\boldsymbol{x}), \quad \gamma(\boldsymbol{x}) = \exp\left(1 - \frac{(3\alpha)^2}{(3\alpha)^2 - \mathrm{dist}(\boldsymbol{x},\mathcal{M})^2}\right), \quad (12)$$

where $\gamma$ is a bump function that is zero outside the $3\alpha$ neighborhood of the manifold, illustrated in Figure 3c. Here $\mathrm{dist}(\boldsymbol{x},\mathcal{M})$ can be computed as the distance from $\boldsymbol{x}$ to its nearest neighbor node $\boldsymbol{x}_j$.[5]

## 4   Efficient Training of the Implicit Manifold Gaussian Processes

Here we describe how to perform the implicit manifold Gaussian process regression efficiently, being able to handle hundreds of thousands of points, in both supervised and semi-supervised regimes.

In all cases we need efficient (approximate) KNN to build a graph, extend the kernel beyond the nodes $\boldsymbol{x}_i$ and combine the geometric model with the standard Euclidean one. For this we use FAISS (Johnson et al., 2019). The resulting sparse matrices, such as the Laplacian $\boldsymbol{\Delta}_{\mathrm{rw}}$, we represent as black box functions capable of performing matrix-vector multiplications for any given input vector.

---

[5]In practice it makes sense to compute $\mathrm{dist}(\boldsymbol{x},\mathcal{M})$ as the average of distances between $\boldsymbol{x}$ and its $K$ nearest neighbors to smoothen the resulting $\gamma(\boldsymbol{x})$—this is computationally cheap given an efficient KNN implementation.

After hyperparameters are found—we will return to their search later—we need to compute the eigenpairs $\lambda_l, \boldsymbol{f}_l$ of $\boldsymbol{\Delta}_{\mathrm{rw}}$. For this we run Lanczos algorithm (Meurant, 2006) to evaluate the eigenpairs $\lambda_l^{\mathrm{sym}}, \boldsymbol{f}_l^{\mathrm{sym}}$ of the symmetric matrix $\boldsymbol{\Delta}_{\mathrm{sym}}$, putting $\lambda_l = \lambda_l^{\mathrm{sym}}$ and $\boldsymbol{f}_l = \mathbf{D}^{-1/2} \boldsymbol{f}_l^{\mathrm{sym}}$ because the matrices $\boldsymbol{\Delta}_{\mathrm{rw}}$ and $\boldsymbol{\Delta}_{\mathrm{sym}}$ are similar, i.e. $\boldsymbol{\Delta}_{\mathrm{rw}} = \mathbf{D}^{-1/2} \boldsymbol{\Delta}_{\mathrm{sym}} \mathbf{D}^{1/2}$. Importantly, Lanczos algorithm only relies on matrix-vector products with $\boldsymbol{\Delta}_{\mathrm{sym}}$. We only compute a few hundred of eigenpairs, asking Lanczos to provide twice or trice as many and disregarding the rest.

When there is a lot of labeled data, we approximate the classical Euclidean (e.g. squared exponential) kernel using random Fourier features approximation (Rahimi and Recht, 2007), this allows linear computational complexity scaling with respect to the number of data points. The number of Fourier features is taken to be equal to $L$, with the same $L$ as in Equation (8).

As it was already mentioned, we need to find hyperparameters $\hat{\boldsymbol{\theta}} = \left( \hat{\alpha}, \hat{\kappa}, \hat{\sigma}^2, \hat{\sigma}_\varepsilon^2 \right)$ that determine the graph, Gaussian process prior and the noise variance that fit the observations $\boldsymbol{y}$ best. The $\nu$ parameter we assume manually fixed. To avoid nonsensical parameter values—a common difficulty often occurring when the data is scarce—one might want to assume a prior $p(\boldsymbol{\theta})$ on $\boldsymbol{\theta}$. Some specific choices of $p(\boldsymbol{\theta})$ are discussed in Appendix C. Then $\hat{\boldsymbol{\theta}}$ is a maximum a posteriori (MAP) estimate:

$$\hat{\boldsymbol{\theta}} = \arg\max_{\boldsymbol{\theta}} \log p(\boldsymbol{y} \mid \boldsymbol{\theta}, \mathbf{X}) + \log p(\boldsymbol{\theta}). \tag{13}$$

To simplify hyperparameter initialization and align with zero prior mean assumption it makes sense to preprocess $y_i$ to be centered and normalized.

To solve the optimization problem in Equation (13) we use restarted gradient descent. Repeatedly evaluating the gradient of $\log p(\boldsymbol{y} \mid \boldsymbol{\theta}, \mathbf{X})$ is the main computational bottleneck. The key idea for doing this efficiently—viable for integer values of $\nu$—is to reduce matrix-vector products with Matérn kernels' precision to iterated matrix-vector products with the Laplacian, which is *sparse*. First, we describe this in detail in the noiseless supervised setting, where the idea is most directly applicable.

## 4.1 Noiseless Supervised Learning

Here we assume that all inputs are labeled, i.e. $N = n$, and all observations are noiseless, i.e. $\sigma_\varepsilon^2 = 0$.

Denoting by $\mathbf{P_{XX}} = \mathbf{K_{XX}^{-1}}$ the precision matrix, the log-likelihood $\log p(\boldsymbol{y} \mid \boldsymbol{\theta}, \mathbf{X})$, up to a multiplicative constant and an additive constant irrelevant for optimization, is given by

$$\mathcal{L}(\boldsymbol{\theta}) = -\log \det(\mathbf{K_{XX}}) - \boldsymbol{y}^\top \mathbf{K_{XX}^{-1}} \boldsymbol{y} = \log \det(\mathbf{P_{XX}}) - \boldsymbol{y}^\top \mathbf{P_{XX}} \boldsymbol{y}. \tag{14}$$

Its gradient may be given and then subsequently approximated (Gardner et al., 2018) by

$$\frac{\partial \mathcal{L}(\boldsymbol{\theta})}{\partial \boldsymbol{\theta}} = \mathrm{tr}\left( \mathbf{P_{XX}^{-1}} \frac{\partial \mathbf{P_{XX}}}{\partial \boldsymbol{\theta}} \right) - \boldsymbol{y}^\top \frac{\partial \mathbf{P_{XX}}}{\partial \boldsymbol{\theta}} \boldsymbol{y} \approx \boldsymbol{z}^\top \mathbf{P_{XX}^{-1}} \frac{\partial \mathbf{P_{XX}}}{\partial \boldsymbol{\theta}} \boldsymbol{z} - \boldsymbol{y}^\top \frac{\partial \mathbf{P_{XX}}}{\partial \boldsymbol{\theta}} \boldsymbol{y}, \tag{15}$$

where $\boldsymbol{z}$ is a random vector consisting of IID variables that are either $1$ or $-1$ with probability $1/2$. The first term on the right-hand side is the stochastic estimate of the trace of Hutchinson (1989). Since the kernels from Section 3.3 coincide with graph Matérn kernels on the nodes $x_i$, we have

$$\mathbf{K_{XX}} = \frac{\sigma^2}{C_{\nu,\kappa}} \sum_{l=0}^{L-1} \Phi_{\nu,\kappa}(\lambda_l) \boldsymbol{f}_l \boldsymbol{f}_l^\top, \qquad \boldsymbol{\Delta}_{\mathrm{rw}} \boldsymbol{f}_l = \lambda_l \boldsymbol{f}_l, \qquad \boldsymbol{f}_l^\top \mathbf{D} \boldsymbol{f}_m = \delta_{lm}. \tag{16}$$

However, as the graph bandwidth $\alpha$ is one of the hyperparameters we optimize over, using Equation (16) would entail repeated eigenpair computations and differentiating through this procedure. Because of this, *we use an alternative way* to compute matrix-vector products $\mathbf{P_{XX}} \boldsymbol{u}$ detailed below.[6]

**Proposition 2.** *Assuming $\nu \in \mathbb{N}$, the precision matrix $\mathbf{P_{XX}}$ of $k_{\nu,\kappa,\sigma^2}^{\mathbf{X}}(\boldsymbol{x}_i, \boldsymbol{x}_j)$ can be given by*

$$\mathbf{P_{XX}} = \frac{\sigma^{-2}}{C_{\nu,\kappa}^{-1}} \mathbf{D} \underbrace{\left( \frac{2\nu}{\kappa^2} \mathbf{I} + \boldsymbol{\Delta}_{\mathrm{rw}} \right) \cdot \ldots \cdot \left( \frac{2\nu}{\kappa^2} \mathbf{I} + \boldsymbol{\Delta}_{\mathrm{rw}} \right)}_{\nu \ times}. \tag{17}$$

*Proof.* See Appendix A. $\qquad\qquad\square$

---

[6]Though automatic differentiability could in principle work for iterative methods like the Lanczos algorithm, the amount of memory required for storing the gradients of the intermediate steps quickly becomes prohibitive.

Using Proposition 2 to evaluate matrix-vector products $\mathbf{P_{XX}}\boldsymbol{u}$ and conjugate gradients (Meurant, 2006) to solve $\boldsymbol{z}^\top \mathbf{P_{XX}^{-1}}$ using only the matrix-vector products, we can efficiently evaluate the right-hand side of Equation (15), with linear costs with respect to $N$, assuming that the graph is sparse. Preconditioning (Wenger et al., 2022) can be used to further improve the efficiency of the solve.

## 4.2 Noiseless Semi-Supervised Learning

Here we assume that inputs are partly unlabeled, i.e. $N \neq n$, while observations are still noiseless, i.e. $\sigma_\varepsilon^2 = 0$. Denote $\mathbf{Z}$ to be the labeled part of $\mathbf{X}$. Then the matrix $\mathbf{K_{XX}}$ in the log-likelihood given by Equation (14) should be substituted with $\mathbf{K_{ZZ}}$. However, while $\mathbf{P_{XX}}$ can be represented using Equation (17), the precision $\mathbf{P_{ZZ}} = \mathbf{K_{ZZ}^{-1}}$ cannot. We thus compute it as the Schur complement:

$$\mathbf{P_{ZZ}} = \mathbf{A} - \mathbf{B}\mathbf{D}^{-1}\mathbf{C}, \qquad\qquad \mathbf{P_{XX}} = \begin{pmatrix} \mathbf{A} & \mathbf{B} \\ \mathbf{C} & \mathbf{D} \end{pmatrix}, \qquad (18)$$

where partitioning of $\mathbf{P_{XX}}$ corresponds to partitioning $\mathbf{X}$ into $\mathbf{Z}$ and the rest. Evaluating a matrix-vector product $\mathbf{P_{ZZ}}\boldsymbol{u}$ requires a solve of $\mathbf{D}^{-1}(\mathbf{C}\boldsymbol{u})$. This solve can also be performed using conjugate gradients, keeping the computational complexity linear in $N$ but increasing the constants.

## 4.3 Handling Noisy Observations

Finally, we assume noisy observations, i.e. $\sigma_\varepsilon^2 > 0$. The inputs can be partially unlabeled, i.e. $N \neq n$.

In this case, matrix $\mathbf{K_{XX}}$ in the log-likelihood given by Equation (14) should be substituted with $\mathbf{K_{ZZ}} + \sigma_\varepsilon^2\mathbf{I}$. To reduce this to the previously considered cases, we use the Taylor expansion

$$(\mathbf{K_{ZZ}} + \sigma_\varepsilon^2\mathbf{I})^{-1} \approx \mathbf{K_{ZZ}^{-1}} - \sigma_\varepsilon^2\mathbf{K_{ZZ}^{-2}} + \sigma_\varepsilon^4\mathbf{K_{ZZ}^{-3}} - \ldots = \mathbf{P_{ZZ}} - \sigma_\varepsilon^2\mathbf{P_{ZZ}^2} + \sigma_\varepsilon^4\mathbf{P_{ZZ}^3} - \ldots \quad (19)$$

In practice, we only use the first two terms on the right-hand side as an approximation. This allows to retain linear computational complexity scaling with respect to $N$ but increases the constants.

## 4.4 Resulting Algorithm

Here we provide a concise summary of the *implicit manifold Gaussian process regression* algorithm.

**Step 1: KNN-index.** Construct the KNN index on the points $\boldsymbol{x}_1, .., \boldsymbol{x}_N$. This allows linear time evaluation of any matrix-vector product with $\tilde{\mathbf{A}}$, and thus also with $\mathbf{A}, \boldsymbol{\Delta}_{rw}, \mathbf{P_{XX}}$ for $\nu \in \mathbb{N}$, etc.

**Step 2: hyperparameter optimization.** Find the hyperparameters $\hat{\boldsymbol{\theta}}$ that solve Equation (13). Assuming $\nu = \hat{\nu} \in \mathbb{N}$ is manually fixed, this relies only on matrix-vector products with $\boldsymbol{\Delta}_{rw}$.

**Step 3: computing the eigenpairs.** Fixing the graph bandwidth $\hat{\alpha}$ found on Step 2, compute the eigenpairs $\lambda_l, f_l$ corresponding to the $L$ smallest eigenvalues $\lambda_l$. For large $N$, use Lanczos algorithm.

After the steps above are finished, Equations (8) and (11) define the geometric kernel $k^{\mathbf{X}}_{\hat{\nu},\hat{\kappa},\hat{\sigma}^2}(\boldsymbol{x}, \boldsymbol{x}')$ for arbitrary $\boldsymbol{x}, \boldsymbol{x}' \in \mathbb{R}^d$. Then the respective prior $\mathrm{GP}(0, k^{\mathbf{X}}_{\hat{\nu},\hat{\kappa},\hat{\sigma}^2})$ can be conditioned by the labeled data in the standard way, yielding the posterior $f^{(m)} \sim \mathrm{GP}(m^{(m)}, k^{(m)})$. To get sensible predictions far away from the data, the geometric model $f^{(m)}$ is convexly combined with an independently trained classical Gaussian process model, as given by Equation (12). The resulting predictive model is still a Gaussian process, sum of two appropriately weighted independent Gaussian processes.

**Remark.** The number of neighbors $K$, the number of eigenpairs $L$ and the smoothness $\nu$ are assumed to be manually fixed parameters. Higher values of $K$ and $L$ improve the quality of approximation of the manifold kernel, which is often linked to better predictive performance, but requires more computational resources. The parameter $\nu$ can be picked using cross validation or prior knowledge. Small integer values of $\nu$ reduce computational costs, but may be inadequate for higher dimensions of the assumed manifold due to the $\nu' = \nu - \dim(\mathcal{M})/2$ link with the manifold kernel smoothness $\nu'$.

# 5 Experiments

We start in Section 5.1 by examining a simple synthetic example to gain intuition on how noise-sensitive the technique is. Then in Section 5.2 we consider real datasets, showing improvements in higher dimensions. More experiments, results, and additional discussion can be found in Appendix B

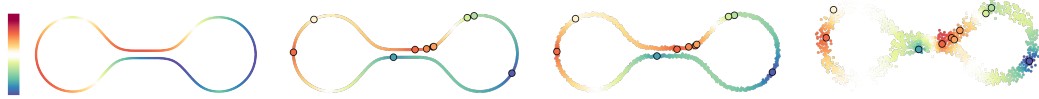

| (a) Ground truth | (b) Noiseless, prediction | (c) Low noise, prediction | (d) High noise, prediction |

Figure 4: The ground truth function on the dumbbell manifold and the predictions of the implicit manifold Gaussian process regression (IMGP) under different levels of noise.

## 5.1 Synthetic Examples

We consider a one dimensional manifold resembling the shape of a *dumbbell* which already appeared in Figures 1 and 3. The unknown function $f_*$ is defined by fixing a point $x^*$ in the top left part of the dumbbell, and computing $\sin(d(x^*, \cdot))$ where $d(\cdot, \cdot)$ denotes the geodesic (intrinsic) distance between a pair of points on the manifold. This function is illustrated in Figure 4a.

To measure performance we primarily rely on measuring negative log-likelihood (NLL) on the dense mesh of test locations. We do this because such metric is able to combine accuracy and calibration simultaneously. Additionally, we present the root mean square error (RMSE).

We investigate a semi-supervised setting where the number of unlabeled points is large ($N-n = 1546$) and the number of labeled points is small ($n = 10$). We contaminate the inputs with noise, putting $\mathbf{X} = \mathbf{X}_{\text{noiseless}} + \mathrm{N}(0, \sigma_{\mathbf{X}}^2 \mathbf{I})$ and do the same with the outputs, putting $\mathbf{y} = f_*(\mathbf{X}) + \mathrm{N}(0, \sigma_{\mathbf{y}}^2 \mathbf{I})$ for various values of $\sigma_{\mathbf{X}}, \sigma_{\mathbf{y}} > 0$. Specifically, we consider $\sigma_{\mathbf{X}} = \sigma_{\mathbf{y}} = \beta \in \{0, 0.01, 0.05\}$ to which we refer to as the noiseless setting, the low noise setting and the high noise setting, respectively.

The results for these are visualized in Figures 4b to 4d with performance metrics reported in Table 1. The implicit manifold Gaussian process regression is referred to as IMGP (we use $\nu = 1$) and it is compared with the standard Euclidean Matérn-$5/2$ Gaussian process. IMGP performs much better in the noiseless and the low noise settings. The high noise is enough to damage the calibration of IMGP, as it ties with the baseline model: NLL is slightly worse and RMSE is slightly better.

In Appendix B.1 we show how performance depends on the fraction $n/N$ of labeled data points, the truncation level $L$ and we discuss the choice of the $K$ parameter in KNN. Additionally, in Appendix B.2 we consider noise-sensitivity for a 2D manifold.

## 5.2 High Dimensional Datasets

For the high-dimensional setting, we considered predicting rotation angles for MNIST-based datasets. Additionally, we examined a high-dimensional dataset from the UCI ML Repository, CT slices.

### 5.2.1 Setup

**Datasets.** We consider two MNIST-based datasets. The first one is created by extracting a single image per digit from the complete MNIST dataset. By randomly rotating these 10 images we obtained $N = 10000$ training samples and 1000 testing samples. We call it *Single Rotated MNIST (SR-MNIST)*. For the second dataset, we select 100 random samples from MNIST. By randomly rotating these, we generate $N = 100\,000$ training samples, most will be unlabeled, and $10\,000$ testing samples. We call it *Multiple Rotated MNIST (MR-MNIST)*. The last dataset, *CT slices*, has dimensionality of $d = 385$, we split it to have $N = 24075$ training samples and $24075$ testing samples. Dataset names can be complemented by the fraction of labeled samples, e.g. MR-MNIST-10% refers to $n = 10\% N$.

| | **RMSE** | | | **NLL** | | |
|---|---|---|---|---|---|---|
| | $\beta = 0$ | $\beta = 0.01$ | $\beta = 0.05$ | $\beta = 0$ | $\beta = 0.01$ | $\beta = 0.05$ |
| Euclidean Matérn-$5/2$ | 0.98 | $0.99 \pm 0.02$ | $1.02 \pm 0.03$ | $-2.17$ | $-2.09 \pm 0.03$ | $\mathbf{-1.91 \pm 0.10}$ |
| IMGP | **0.33** | $\mathbf{0.34 \pm 0.02}$ | $\mathbf{1.00 \pm 0.02}$ | $\mathbf{-5.02}$ | $\mathbf{-4.19 \pm 0.1}$ | $-1.91 \pm 1.88$ |

Table 1: Performance metrics for the dumbbell manifold with varying magnitude of noise $\beta$.

|         | MNIST | | | CT slices | | |
| Method | SR - 10% | MR - 1% | MR - 10% | 5% | 10% | 25% |
|---|---|---|---|---|---|---|
| EGP | $-0.54 \pm 0.01$ | $-0.20 \pm 0.01$ | $-0.43 \pm 0.01$ | $-0.80 \pm 0.02$ | $-0.96 \pm 0.00$ | $-1.20 \pm 0.09$ |
| S-IMGP | $-1.42 \pm 0.01$ | $2.24 \pm 0.20$ | $-0.68 \pm 0.08$ | $0.47 \pm 0.06$ | $-0.59 \pm 0.08$ | $-0.08 \pm 0.01$ |
| SS-IMGP | $\mathbf{-1.52 \pm 0.01}$ | $\mathbf{-0.59 \pm 0.01}$ | $\mathbf{-0.79 \pm 0.00}$ | $26.1 \pm 12.7$ | $1.03 \pm 0.09$ | $-0.72 \pm 0.68$ |
| S-IMGP (full) | - | - | - | $0.64 \pm 0.83$ | $0.88 \pm 0.29$ | $-0.42 \pm 0.10$ |
| SS-IMGP (full) | - | - | - | $\mathbf{-2.48 \pm 0.08}$ | $\mathbf{-2.35 \pm 0.04}$ | $\mathbf{-1.99 \pm 0.04}$ |

Table 2: Negative log likelihood on test samples for real datasets. For RMSE see Tables 4 and 5.

**Methods.** We consider implicit manifold Gaussian processes in the supervised regime (*S-IMGP*) and in the semi-supervised regime (*SS-IMGP*). We compare them to the GPyTorch implementation of the Euclidean Matérn-$5/2$ Gaussian Process. We refer to it as the *Euclidean Gaussian Process (EGP)*.

**Additional details.** We run 100 iterations of hyperparameter optimization using Adam with a fixed learning rate of $0.01$. For MNIST, with use IMGP with $\nu = 2$; for CT slices—with $\nu = 3$.

### 5.2.2 Results

Table 2 shows the negative log-likelihood metric for different datasets and methods on the test set. The respective RMSEs are presented in Appendix B.3. On SR-MNIST, IMGP outperforms EGP in both supervised and semi-supervised scenarios. MR-MNIST is more challenging. In the supervised setting for $n = 1\%N$, S-IMGP is incapable of inferring the underlying manifold structure, performing worse than EGP. However, SS-IMGP, with more data to infer manifold from, performs best. For $n = 10\%N$, IMGP gets a better grip of the dataset's geometry, outperforming EGP in both regimes.

For CT slices, regardless of $n$, both S-IMGP and SS-IMGP performed poorly. Looking for an explanation, we considered two modifications. First, we fixed the graph bandwidth $\hat{\alpha}$ found in the algorithm's Step 2 (cf. Section 4.4), and re-optimized the other hyperparameters $\kappa, \sigma^2, \sigma_\varepsilon^2$ by maximizing the likelihood of the eigenpair-based model (truncated to $L = 2000$ eigenpairs) computed in the algorithm's Step 3. This resulted in limited improvement but did not change the big picture—in fact, values for S-IMGP and SS-IMGP in Table 2 for CT slices already include this modification.

Second, on top of this hyperparameter re-optimization, we tried computing the eigenpairs using torch.linalg.eigh instead of the Lanczos implementation in GPyTorch, taking the same number $L = 2000$ of eigenpairs. The resulting methods S-IMGP (full) and SS-IMGP (full) showed considerable improvement over the baseline, as shown in Table 2. This indicated an issue with the quality of eigenpairs derived from the Lanczos method which requires further investigation. We discuss this in Appendix B.3, together with the aforementioned hyperparameter re-optimization procedure.

## 6   Conclusion

In this work, we propose the *implicit manifold Gaussian process regression* technique. It is able to use unlabeled data to improve predictions and uncertainty calibration by learning the implicit manifold upon which the data lies, being inspired by the convergence of graph Matérn Gaussian processes to their manifold counterparts. This helps building better probabilistic models in higher dimensional settings where the standard Euclidean Gaussian processes usually struggle. This is supported by our experiments in a synthetic low-dimensional setting and for high-dimensional datasets. Leveraging sparse structure of graph Matérn precision matrices and efficient approximate KNN, the technique is able to scale to large datasets of hundreds of thousands points, which is especially important in high dimension, where a large number of unlabeled points is often needed to learn the implicit manifold. The model is fully differentiable, making it possible to infer hyperparameters in the usual way.

**Limitations.**   The quality of the constructed graph significantly influences the technique's performance. When dealing with data from complex manifolds or exhibiting highly non-uniform density, simplistic KNN strategies might fail to capture the manifold structure due to their reliance on a single graph bandwidth. In such scenarios, larger values of parameters $K$ and $L$, or in high dimensions, of parameter $\nu$, may be beneficial but could substantially increase computational costs. Furthermore, larger datasets coupled with high parameter values can lead to numerical stability issues, for instance, in the Lanczos algorithm, calling for further improvements and research. Despite these challenges, our method shows promise for advancing probabilistic modeling in higher dimensions.

## Acknowledgements

We express our gratitude to Alexander Shulzhenko (St. Petersburg University, mentored by VB), whose initial work on a similar problem and accessible source material at `https://github.com/AlexanderShulzhenko/Implicit-Manifold-Gaussian-Processes`, while not included in the paper, sparked inspiration for this project. We thank Dr. Alexander Terenin (Cornell University) for generously making his Blender rendering scripts accessible to the public. These scripts, which aided us in creating Figure 2, can be found in his Ph.D. thesis repository at `https://github.com/aterenin/phdthesis`. BF and AB acknowledge support by the European Research Council (ERC), Advanced Grant agreement No 741945, Skill Acquisition in Humans and Robots. VB acknowledges support by an ETH Zürich Postdoctoral Fellowship.

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

# A   Theory

**Proposition 1.** *Denote the eigenpairs by $\lambda_l, f_l$ for a graph Laplacian and by $\lambda_l^{\mathcal{M}}, f_l^{\mathcal{M}}$ for the Laplace–Beltrami operator. Fix $\delta > 0$. Assume that, with probability at least $1 - \delta$, for all $\varepsilon > 0$, for $\alpha$ small enough and for $N$ large enough we have $|\lambda_l - \lambda_l^{\mathcal{M}}| < \varepsilon$ and $|f_l(\boldsymbol{x}_i) - f_l^{\mathcal{M}}(\boldsymbol{x}_i)| < \varepsilon$. Then, with probability at least $1 - \delta$, we have $k_{\nu,\kappa,\sigma^2}^{N,\alpha,L}(\boldsymbol{x}_i, \boldsymbol{x}_j) \to k_{\nu,\kappa,\sigma^2}(\boldsymbol{x}_i, \boldsymbol{x}_j)$ as $\alpha \to 0$, $N, L \to \infty$, where*

$$k_{\nu,\kappa,\sigma^2}^{N,\alpha,L}(\boldsymbol{x}_i, \boldsymbol{x}_j) = \frac{\sigma^2}{C_{\nu,\kappa}} \sum_{l=0}^{L-1} \left(\frac{2\nu}{\kappa^2} + \lambda_l\right)^{-\nu-\dim(\mathcal{M})/2} f_l(\boldsymbol{x}_i) f_l(\boldsymbol{x}_j). \tag{6}$$

*Proof.* Fix small $\varepsilon > 0$. We will prove that for $\alpha$ small enough and $N, L$ large enough we have $|k_{\nu,\kappa,\sigma_f^2}(\boldsymbol{x}_i, \boldsymbol{x}_j) - k_{\nu,\kappa,\sigma_f^2}^{N,\alpha,L}(\boldsymbol{x}_i, \boldsymbol{x}_j)| < C\varepsilon$ for some $C > 0$ with probability at least $1 - \delta$. Since the assumption holds on the same event of probability $1 - \delta$ for all $\varepsilon$, this directly translates to the convergence on the same event. In fact, a probabilistic narrative is nonessential for what we actually prove, and we do not use it below. To simplify notation, we replace $\sum_{l=0}^{L-1}$ by $\sum_{l=0}^{L}$.

First, for any $L \in \mathbb{Z}_{>0}$ define the truncated version $k_{\nu,\kappa,\sigma_f^2}^{L}$ of the manifold kernel $k_{\nu,\kappa,\sigma_f^2}$ by

$$k_{\nu,\kappa,\sigma_f^2}^{L}(\boldsymbol{x}, \boldsymbol{x}') = \frac{\sigma_f^2}{C_{\nu,\kappa}} \sum_{l=0}^{L} \left(\frac{2\nu}{\kappa^2} + \lambda_l^{\mathcal{M}}\right)^{-\nu-\dim(\mathcal{M})/2} f_l^{\mathcal{M}}(\boldsymbol{x}) f_l^{\mathcal{M}}(\boldsymbol{x}'). \tag{20}$$

The manifold Matérn kernels are the reproducing kernels of Sobolev spaces, if the latter are defined appropriately (Borovitskiy et al., 2020). These are Mercer kernels (De Vito et al., 2021), hence, by Mercer's theorem, $k_{\nu,\kappa,\sigma_f^2}^{L} \to k_{\nu,\kappa,\sigma_f^2}$ uniformly on $\mathcal{M}$, i.e. for $L$ large enough we have

$$\left|k_{\nu,\kappa,\sigma_f^2}^{L}(\boldsymbol{x}_i, \boldsymbol{x}_j) - k_{\nu,\kappa,\sigma_f^2}(\boldsymbol{x}_i, \boldsymbol{x}_j)\right| \leq \sup_{\boldsymbol{x},\boldsymbol{x}' \in \mathcal{M}} \left|k_{\nu,\kappa,\sigma_f^2}^{L}(\boldsymbol{x}, \boldsymbol{x}') - k_{\nu,\kappa,\sigma_f^2}(\boldsymbol{x}, \boldsymbol{x}')\right| < \varepsilon. \tag{21}$$

Now suppose that $\alpha$ is small enough and $N$ is large enough so that

$$\left|\lambda_l - \lambda_l^{\mathcal{M}}\right| < \varepsilon', \quad \left|f_l(\boldsymbol{x}_i) - f_l^{\mathcal{M}}(\boldsymbol{x}_i)\right| < \varepsilon', \quad \varepsilon' = \min\left(1, \left(\lambda_L^{\mathcal{M}}\right)^{-\frac{d-1}{4}}, \left(\lambda_L^{\mathcal{M}}\right)^{-\frac{d-1}{2}}\right)\frac{\varepsilon}{L} \tag{22}$$

for all $l \in \{1, .., L\}$ and for all $i \in \{1, .., N\}$ with probability at least $1 - \delta$.

Assuming the manifold is connected, by Donnelly (2006) we have $|f_l^{\mathcal{M}}| \leq C_\lambda \left(\lambda_l^{\mathcal{M}}\right)^{\frac{d-1}{4}}$ for $l > 0$, where $C_\lambda > 0$ is a constant that depends on the geometry of the manifold. The case $l = 0$ is special because $\lambda_0^{\mathcal{M}} = 0$. Since $f_0^{\mathcal{M}}$ is a constant function, we have

$$|f_l^{\mathcal{M}}| \leq C_\lambda \max((\lambda_l^{\mathcal{M}})^{\frac{d-1}{4}}, 1) \tag{23}$$

for all $l \geq 0$, where $C_\lambda$ here is potentially different from the $C_\lambda$ before. Assuming, without loss of generality, $\varepsilon < 1$, we have

$$\left|f_l(\boldsymbol{x}_i) f_l(\boldsymbol{x}_j) - f_l^{\mathcal{M}}(\boldsymbol{x}_i) f_l^{\mathcal{M}}(\boldsymbol{x}_j)\right| \leq |f_l(\boldsymbol{x}_i)| \left|f_l(\boldsymbol{x}_j) - f_l^{\mathcal{M}}(\boldsymbol{x}_j)\right| \tag{24}$$

$$+ \left|f_l^{\mathcal{M}}(\boldsymbol{x}_j)\right| \left|f_l(\boldsymbol{x}_i) - f_l^{\mathcal{M}}(\boldsymbol{x}_i)\right| \tag{25}$$

$$\leq \varepsilon' \cdot \left(|f_l(\boldsymbol{x}_i)| + \left|f_l^{\mathcal{M}}(\boldsymbol{x}_j)\right|\right) \tag{26}$$

$$\leq \varepsilon' \cdot \left(\left|f_l(\boldsymbol{x}_i) - f_l^{\mathcal{M}}(\boldsymbol{x}_i)\right| + \left|f_l^{\mathcal{M}}(\boldsymbol{x}_i)\right| + \left|f_l^{\mathcal{M}}(\boldsymbol{x}_j)\right|\right) \tag{27}$$

$$\leq \varepsilon' \cdot \left(\varepsilon' + 2C_\lambda \max((\lambda_l^{\mathcal{M}})^{\frac{d-1}{4}}, 1)\right) \leq \frac{(1 + 2C_\lambda)\varepsilon}{L}. \tag{28}$$

The function $\Phi(\lambda) = \left(\frac{2\nu}{\kappa^2} + \lambda\right)^{-\nu-\dim(\mathcal{M})/2}$ is Lipschitz: $|\Phi(\lambda) - \Phi(\lambda')| \leq C_\Phi |\lambda - \lambda'|$ where

$$C_\Phi = \sup_{\lambda \geq 0} |\Phi'(\lambda)| = \sup_{\lambda \geq 0} (\nu + \dim(\mathcal{M})/2)\left(\frac{2\nu}{\kappa^2} + \lambda\right)^{-\nu-\dim(\mathcal{M})/2-1} = (\nu + \dim(\mathcal{M})/2)\left(\frac{2\nu}{\kappa^2}\right)^{-\nu-\dim(\mathcal{M})/2-1}. \tag{29}$$

Define an auxiliary kernel with manifold eigenvalues and graph eigenfunctions by

$$\tilde{k}^L_{\nu,\kappa,\sigma^2_f}(\boldsymbol{x},\boldsymbol{x}') = \frac{\sigma^2_f}{C_{\nu,\kappa}} \sum_{l=0}^{L} \left(\frac{2\nu}{\kappa^2} + \lambda_l^{\mathcal{M}}\right)^{-\nu-\dim(\mathcal{M})/2} f_l(\boldsymbol{x}) f_l(\boldsymbol{x}'). \tag{30}$$

Then

$$\frac{C_{\nu,\kappa}}{\sigma^2_f} \left| k^L_{\nu,\kappa,\sigma^2_f}(\boldsymbol{x}_i,\boldsymbol{x}_j) - \tilde{k}^L_{\nu,\kappa,\sigma^2_f}(\boldsymbol{x}_i,\boldsymbol{x}_j) \right| \leq \sum_{l=0}^{L} \left(\frac{2\nu}{\kappa^2} + \lambda_l^{\mathcal{M}}\right)^{-\nu-\dim(\mathcal{M})/2} \frac{(1+2C_\lambda)\varepsilon}{L} \tag{31}$$

$$\leq \Phi(0)(1+2C_\lambda)\varepsilon. \tag{32}$$

Also, noting that $|f_l(\boldsymbol{x}_i)| \leq \left|f_l^{\mathcal{M}}(\boldsymbol{x}_i) - f_l(\boldsymbol{x}_i)\right| + \left|f_l^{\mathcal{M}}(\boldsymbol{x}_i)\right| \leq \varepsilon' + C_\lambda \max((\lambda_l^{\mathcal{M}})^{\frac{d-1}{4}}, 1)$, write

$$\frac{C_{\nu,\kappa}}{\sigma^2_f} \left| \tilde{k}^L_{\nu,\kappa,\sigma^2_f}(\boldsymbol{x}_i,\boldsymbol{x}_j) - k^{N,\alpha,L}_{\nu,\kappa,\sigma^2_f}(\boldsymbol{x}_i,\boldsymbol{x}_j) \right| \leq \sum_{l=0}^{L} \left|\Phi(\lambda_l^{\mathcal{M}}) - \Phi(\lambda_l)\right| |f_l(\boldsymbol{x}_i)| |f_l(\boldsymbol{x}_j)| \tag{33}$$

$$\leq \sum_{l=0}^{L} C_\Phi \left|\lambda_l^{\mathcal{M}} - \lambda_l\right| |f_l(\boldsymbol{x}_i)| |f_l(\boldsymbol{x}_j)| \tag{34}$$

$$\leq \sum_{l=0}^{L} 2C_\Phi \varepsilon' \left((\varepsilon')^2 + C_\lambda^2 \max((\lambda_l^{\mathcal{M}})^{\frac{d-1}{2}}, 1)\right) \tag{35}$$

$$\leq 2C_\Phi(1+C_\lambda^2)\varepsilon. \tag{36}$$

Finally,

$$\left|k_{\nu,\kappa,\sigma^2_f}(\boldsymbol{x}_i,\boldsymbol{x}_j) - k^{N,\alpha,L}_{\nu,\kappa,\sigma^2_f}(\boldsymbol{x}_i,\boldsymbol{x}_j)\right| \leq \left|k_{\nu,\kappa,\sigma^2_f}(\boldsymbol{x}_i,\boldsymbol{x}_j) - k^L_{\nu,\kappa,\sigma^2_f}(\boldsymbol{x}_i,\boldsymbol{x}_j)\right| \tag{37}$$

$$+ \left|k^L_{\nu,\kappa,\sigma^2_f}(\boldsymbol{x}_i,\boldsymbol{x}_j) - \tilde{k}^L_{\nu,\kappa,\sigma^2_f}(\boldsymbol{x}_i,\boldsymbol{x}_j)\right| \tag{38}$$

$$+ \left|\tilde{k}^L_{\nu,\kappa,\sigma^2_f}(\boldsymbol{x}_i,\boldsymbol{x}_j) - k^{N,\alpha,L}_{\nu,\kappa,\sigma^2_f}(\boldsymbol{x}_i,\boldsymbol{x}_j)\right| \tag{39}$$

$$\leq \varepsilon + \frac{\sigma^2_f}{C_{\nu,\kappa}} \left(\Phi(0)(1+2C_\lambda) + 2C_\Phi(1+C_\lambda^2)\right)\varepsilon. \tag{40}$$

This proves the claim. $\square$

**Proposition 2.** *Assuming $\nu \in \mathbb{N}$, the precision matrix $\mathbf{P_{XX}}$ of $k^{\mathbf{X}}_{\nu,\kappa,\sigma^2}(\boldsymbol{x}_i,\boldsymbol{x}_j)$ can be given by*

$$\mathbf{P_{XX}} = \frac{\sigma^{-2}}{C_{\nu,\kappa}^{-1}} \mathbf{D} \underbrace{\left(\frac{2\nu}{\kappa^2}\mathbf{I} + \boldsymbol{\Delta}_{\mathrm{rw}}\right) \cdot \ldots \cdot \left(\frac{2\nu}{\kappa^2}\mathbf{I} + \boldsymbol{\Delta}_{\mathrm{rw}}\right)}_{\nu \text{ times}}. \tag{17}$$

*Proof.* The covariance matrix $\mathbf{K_{XX}}$ is given by

$$\mathbf{K_{XX}} = \frac{\sigma^2}{C_{\nu,\kappa}} \sum_{l=0}^{L-1} \Phi(\lambda_l) \boldsymbol{f}_l \boldsymbol{f}_l^\top, \qquad \Phi(\lambda) = \left(\frac{2\nu}{\kappa^2} + \lambda\right)^{-\nu} \tag{41}$$

where $\boldsymbol{f}_l = \mathbf{D}^{-1/2} \boldsymbol{f}_l^{\mathrm{sym}}$ and $\boldsymbol{f}_l^{\mathrm{sym}}$ are the orthonormal eigenvectors of the symmetric normalized Laplacian $\boldsymbol{\Delta}_{\mathrm{sym}}$. Denote

$$\mathbf{K_{XX}^{sym}} = \frac{\sigma^2}{C_{\nu,\kappa}} \sum_{l=0}^{L-1} \Phi(\lambda_l) \boldsymbol{f}_l^{\mathrm{sym}} (\boldsymbol{f}_l^{\mathrm{sym}})^\top \tag{42}$$

then $(\mathbf{K_{XX}^{sym}})^{-1} = \frac{\sigma^{-2}}{C_{\nu,\kappa}^{-1}} \sum_{l=0}^{L-1} \left(\frac{2\nu}{\kappa^2} + \lambda_l\right)^\nu \boldsymbol{f}_l^{\mathrm{sym}} (\boldsymbol{f}_l^{\mathrm{sym}})^\top = \frac{\sigma^{-2}}{C_{\nu,\kappa}^{-1}} \left(\frac{2\nu}{\kappa^2}\mathbf{I} + \boldsymbol{\Delta}_{\mathrm{sym}}\right)^\nu$. We have

$$\mathbf{K_{XX}} = \frac{\sigma^2}{C_{\nu,\kappa}} \sum_{l=0}^{L-1} \Phi(\lambda_l) \mathbf{D}^{-1/2} \boldsymbol{f}_l^{\mathrm{sym}} (\boldsymbol{f}_l^{\mathrm{sym}})^\top \mathbf{D}^{-1/2} = \mathbf{D}^{-1/2} \mathbf{K_{XX}^{sym}} \mathbf{D}^{-1/2}. \tag{43}$$

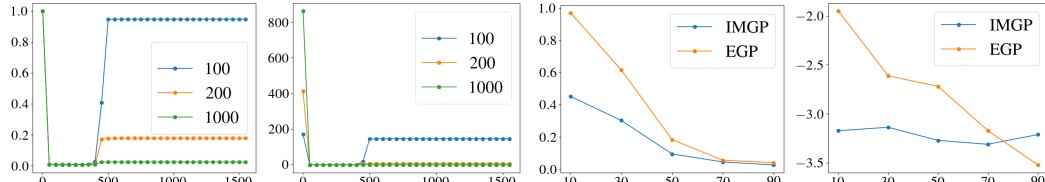

(a) RMSE depending on $L$  (b) NLL depending on $L$  (c) RMSE depending on $f$  (d) NLL depending on $f$

Figure 5: Root Mean Square Error (RMSE) and Negative Log-Likelihood (NLL) for increasing number of eigenpairs $L$ (left panels) and increasing fraction $f = n\%N$ of labeled points (right panels). The legend in (a) and (b) refers to the number of hyperparameter optimization iterations.

On the other hand,

$$\frac{\sigma_f^{-2}}{C_{\nu,\kappa}^{-1}}\mathbf{D}\left(\frac{2\nu}{\kappa^2}\mathbf{I} + \boldsymbol{\Delta}_{\mathrm{rw}}\right)^\nu = \frac{\sigma_f^{-2}}{C_{\nu,\kappa}^{-1}}\mathbf{D}\left(\frac{2\nu}{\kappa^2}\mathbf{I} + \mathbf{D}^{-1/2}\boldsymbol{\Delta}_{\mathrm{sym}}\mathbf{D}^{1/2}\right)^\nu \tag{44}$$

$$= \frac{\sigma_f^{-2}}{C_{\nu,\kappa}^{-1}}\mathbf{D}^{1/2}\left(\frac{2\nu}{\kappa^2}\mathbf{I} + \boldsymbol{\Delta}_{\mathrm{sym}}\right)^\nu \mathbf{D}^{1/2} = \mathbf{D}^{1/2}(\mathbf{K}_{\mathbf{XX}}^{\mathrm{sym}})^{-1}\mathbf{D}^{1/2} \tag{45}$$

$$= \mathbf{K}_{\mathbf{XX}}^{-1} = \mathbf{P}_{\mathbf{XX}}. \tag{46}$$

$\square$

# B  Additional Experimental Results and Details

Here we provide additional results and details for synthetic examples and high-dimensional datasets.

## B.1  1D Dumbbell Manifold

In this section, we analyze our method's sensitivity to the number of labeled points, spectrum truncation, and neighbor count in graph construction, offering deeper insights into its behavior.

**Eigenpairs Truncation.** From a theoretical standpoint, utilizing the complete set of eigenpairs to construct the kernel should be beneficial. However, this assumption may not hold true in cases where the optimization problem is not fully converged. The trends of RMSE and NLL, as depicted in Figures 5a and 5b, illustrate the impact of increasing the number of eigenpairs for 100, 200, and 1000 iterations. In situations where hyperparameter optimization does not converge fully, the length scale parameter $\kappa$ fails to reach sufficiently high values necessary to generate appropriate spectral density decay, which in turn would properly weigh higher frequency eigenpairs. In such scenarios, truncating the spectrum is similar to increasing the length scale, which might improve results in certain cases. In the 1D scenario, due to its simplicity, we opted for a modest number of eigenpairs, namely $L = 50$. Exceeding this count did not yield any discernible improvements.

**Dataset Size.** We analyze the sensitivity to dataset size of our method (IMGP) against the Euclidean case (EGP) in the semi-supervised learning scenario. For fixed number of eigenpairs, $L = 50$, Figures 5c and 5d show performance metrics (RMSE and NLL) depending on the percentage $n\%N$ of labeled points. In a typical scenario where we have at our disposal fewer labeled points compared to the number of unlabeled points, our method outperforms EGP in both accuracy (RMSE) and uncertainty quantification (NLL). As $n$ increases EGP starts to match the performance of IMGP.

**Number of neighbors.** For the one dimensional case considered in this section, only three neighbors are necessary to capture the essential features of the manifold's geometry. Choosing a number less than three would hinder the algorithm's ability to capture the underlying manifold structure. On the other hand, increasing the number of neighbors beyond this threshold does not affect the solution, provided that sufficient time is allocated for hyperparameter optimization to converge and the graph bandwidth becomes small enough to "correct" for all undesired edges in the graph structure (for instance in the central region of the dumbbell). In high-dimensional problems where the dimension of the underlying manifold is unknown, this suggests to incrementally increase the number of neighbors until the loss function stops improving, if it is computationally feasible to try multiple values of $K$.

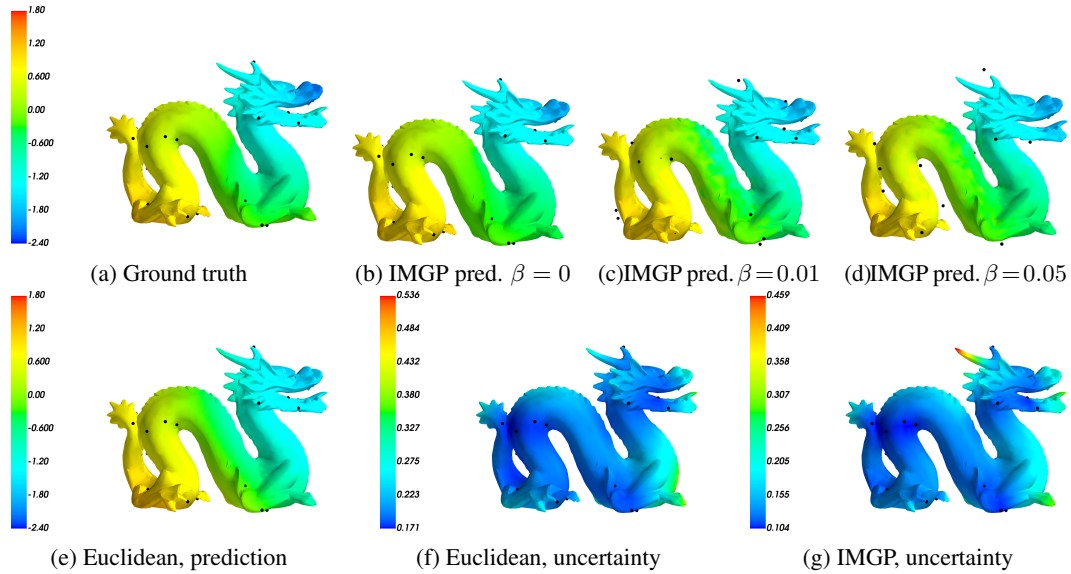

(a) Ground truth     (b) IMGP pred. $\beta = 0$    (c) IMGP pred. $\beta = 0.01$    (d) IMGP pred. $\beta = 0.05$

(e) Euclidean, prediction      (f) Euclidean, uncertainty      (g) IMGP, uncertainty

Figure 6: (a) Ground truth function on the complex 2D manifold and (b)-(d) predictions of the implicit manifold Gaussian process regression (IMGP) for increasing level of sampling noise $\beta$. (e)-(d) Euclidean GP prediction and uncertainty and (g) IMGP uncertainty in noiseless scenario.

## B.2    2D Dragon Manifold

We consider another synthetic setting, a complex 2D manifold, depicted in Figure 6. The ground truth function is visualized in Figure 6a, it is the same as in the one-dimensional case, the sine of the geodesic distance to a point, which is located in the green area.

In the semi-supervised learning scenario, Figures 6e and 6b offer a comparison of the posterior mean between the Euclidean GP and IMGP, while Figures 6f and 6g illustrate the posterior standard deviation for both models—quite similar to each other, in this regime. Similar to what we did for the 1D compact manifold in Section 5.1, we evaluated the performance of IMGP under varying levels of sampling noise. Figures 6b to 6d display the IMGP predictions in the semi-supervised learning scenario as sampling noise increases. Similar to the 1D case, our approach consistently outperforms the standard Euclidean GP, as evidenced in Table 3.

**Remark.** We observed that for higher levels of sampling noise, the linear combination of posteriors, as described by Equation (12), significantly outperform the single geometric model.

## B.3    High Dimensional Datasets

**Rotated MNIST.** For IMGP, we use $\nu = 2$. As discussed in Appendix B.1, the optimal number of eigenpairs $L$ varies considerably depending on the convergence of the optimization problem. Given the limited number of iterations per run we opted to fix the number of eigenpairs at $20\% N$ and $2\% N$, for SR-MNIST and MR-MNIST, respectively. Table 4 reports the complete results obtained for the rotated MNIST dataset, including both RMSE and NLL. Notably, these emphasize the importance of unlabeled points, as S-IMGP performs worse than both SS-IMGP and EGP on MR-MNIST-1%.

| | RMSE | | | NLL | | |
|---|---|---|---|---|---|---|
| | $\beta = 0$ | $\beta = 0.01$ | $\beta = 0.05$ | $\beta = 0$ | $\beta = 0.01$ | $\beta = 0.05$ |
| Euclidean Matérn-$5/2$ | $0.24 \pm 0.02$ | $0.24 \pm 0.01$ | $0.26 \pm 0.00$ | $-0.85 \pm 0.46$ | $0.16 \pm 1.58$ | $1.26 \pm 1.80$ |
| IMGP | $\mathbf{0.12 \pm 0.01}$ | $\mathbf{0.21 \pm 0.01}$ | $\mathbf{0.22 \pm 0.01}$ | $\mathbf{-2.14 \pm 0.13}$ | $\mathbf{-1.51 \pm 0.03}$ | $\mathbf{-1.30 \pm 0.09}$ |

Table 3: Results for a complex 2D manifold with varying magnitude of sampling noise.

**CT slices.** For IMGP, we use $\nu = 3$. In Table 5 we report results for IMGP-S (full) and IMGP-SS (full). These are based on the "exact" eigenpairs, computed by torch.linalg.eigh, as opposed to the standard Lanczos implementation we use by default. Additionally, these include the hyperparameter re-optimization step, as described in Section 5.2 and discussed below. Regarding RMSE, all three compared methods—IMGP-S (full) and IMGP-SS (full) and EGP—exhibit similar performance, with a slight advantage for SS-IMGP (full) as the number of training points increases. When considering NLL, as previously observed with MNIST, SS-IMGP performs best in all settings. However, NLL decreases for larger values of $n/N$, indicating a probable overfit in this regime.

**Hyperparameter re-optimization.** We observed this step to serve two important purposes: (1) fixing overly small values of the signal variance $\sigma^2$, potentially caused by the absence of covariance normalization in the optimization process and poor convergence; (2) adjusting the length scale parameter to take into account the loss of high-frequency components due to truncation.

**Implementation.** Currently, the implementation faces two significant limitations that might restrict its usability to high-memory hardware setups. Firstly, due to the absence of rich enough sparse matrix routines in PyTorch, we had to develop our own custom implementation of differentiable sparse operators. We kept to high-level routines, which forced us to strike a balance between performance and memory efficiency. In particular, for matrix-vector sparse product operations, our approach relies on highly optimized vectorized code, delivering high performance on GPU at the expense of increased memory allocation. Secondly, we faced challenges with sparse eigen-solvers in the PyTorch ecosystem. Our attempts of using the PyTorch implementation of the LOBPCG algorithm, torch.lobpcg, yielded relatively poor results. We had similar experience with the Lanczos implementation from GPyTorch (Gardner et al., 2018). In light of this, when extracting higher-frequency components was needed, we resorted to two alternative solutions: PyTorch dense matrix eigen-decomposition torch.linalg.eigh and the SciPy Arpack wrapper scipy.linalg.eigsh. The first approach, while benefiting from GPU acceleration, can be infeasible because of limited GPU memory. The second approach, known for its efficiency in memory usage due to its Krylov subspace-based nature, is constrained to CPU utilization, significantly impacting the algorithm's overall performance.

## C Hyperparameter Priors and Initialization

Here we describe hyperpameter priors which might be of help when using implicit manifold Gaussian process regression.

**Graph Bandwidth $\alpha$.** Graph Laplacian converges to the Laplace–Beltrami operator when $\alpha$ tends to zero, motivating smaller $\alpha$. However, in a non-asymptotic setting it is impractical to have $\alpha$ overly small as it will render the graph effectively disconnected and cause numerical instabilities. One appropriate prior could thus be *gamma distribution*, whose right tail discourages high values of $\alpha$, and which, given an appropriate choice of the parameters, encourages $\alpha$ to align with the scale of pairwise distances between graph nodes. Specifically, we choose the parameters of the gamma distribution so as to (1) match its mode with the median ($Q_2$) of pairwise average distance between $K$-th nearest neighbors because such an $\alpha$ would give reasonable weights in the KNN graph and (2) so as to place its standard $0.95$ confidence interval to the right of a certain lower bound $\bar{\alpha}$ we define further that ensures the graph is numerically not disconnected. Let

$$\mathcal{D} = \left\{ \max_{\boldsymbol{x}_i \in \mathrm{KNN}(\boldsymbol{x}_j)} \|\boldsymbol{x}_i - \boldsymbol{x}_j\| \text{ where } j = 1, .., N \right\}. \tag{47}$$

| Dataset | $n/N$ | RMSE | | | NLL | | |
|---------|-------|--------|---------|--------|--------|---------|--------|
| | | **S-IMGP** | **SS-IMGP** | **EGP** | **S-IMGP** | **SS-IMGP** | **EGP** |
| SR-MNIST | 10% | $0.04 \pm 0.01$ | $\mathbf{0.01 \pm 0.00}$ | $0.12 \pm 0.01$ | $-1.42 \pm 0.01$ | $\mathbf{-1.52 \pm 0.01}$ | $-0.54 \pm 0.01$ |
| MR-MNIST | 1% | $0.74 \pm 0.02$ | $\mathbf{0.43 \pm 0.02}$ | $0.74 \pm 0.02$ | $2.24 \pm 0.20$ | $\mathbf{-0.59 \pm 0.01}$ | $-0.20 \pm 0.01$ |
| MR-MNIST | 10% | $0.14 \pm 0.05$ | $\mathbf{0.03 \pm 0.00}$ | $0.13 \pm 0.00$ | $-0.68 \pm 0.08$ | $\mathbf{-0.79 \pm 0.00}$ | $-0.43 \pm 0.01$ |

Table 4: Results for the rotated MNIST dataset.

| | RMSE | | | NLL | | |
|---|---|---|---|---|---|---|
| $n/N$ | **S-IMGP (full)** | **SS-IMGP (full)** | **EGP** | **S-IMGP (full)** | **SS-IMGP (full)** | **EGP** |
| 5% | $0.27 \pm 0.02$ | $0.39 \pm 0.04$ | $\mathbf{0.24 \pm 0.02}$ | $0.64 \pm 0.83$ | $\mathbf{-2.48 \pm 0.08}$ | $-0.80 \pm 0.02$ |
| 10% | $0.22 \pm 0.02$ | $0.19 \pm 0.01$ | $\mathbf{0.16 \pm 0.00}$ | $0.88 \pm 0.29$ | $\mathbf{-2.35 \pm 0.04}$ | $-0.96 \pm 0.00$ |
| 25% | $0.14 \pm 0.01$ | $\mathbf{0.08 \pm 0.02}$ | $0.09 \pm 0.01$ | $-0.42 \pm 0.10$ | $\mathbf{-1.99 \pm 0.04}$ | $-1.20 \pm 0.09$ |
| 50% | $0.08 \pm 0.01$ | $\mathbf{0.07 \pm 0.01}$ | $0.08 \pm 0.04$ | $-0.96 \pm 0.11$ | $\mathbf{-2.04 \pm 0.02}$ | $-1.02 \pm 0.04$ |

Table 5: Results for Relative location of CT slices on axial axis ($d = 385$, $N = 48150$) from UCI Machine Learning Repository.

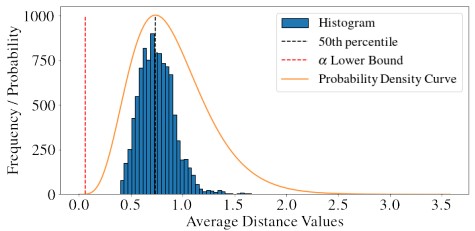

(a) Prior for the semi-supervised case

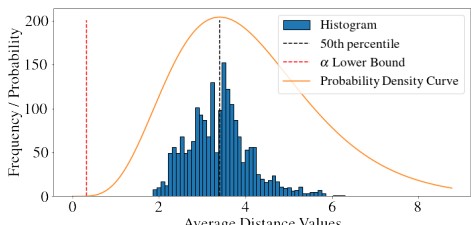

(b) Prior for the supervised case

Figure 7: The histogram of $\mathcal{D}$ and the prior for the bandwidth hyperparameter $\alpha$.

The bandwidth's lower bound we compute as

$$\bar{\alpha} = \min_{d \in \mathcal{D}} \sqrt{-\frac{d^2}{4 \log \tau}}, \tag{48}$$

where $\tau$ is a user-defined parameter.

Let $\eta$ and $\beta$ the shape and rate parameters of the gamma distribution. In order to achieve (1), we define

$$\eta = \rho Q_2 + 1 \quad \text{and} \quad \beta = \rho \tag{49}$$

where $\rho$ is used to achieve (2) and is given by

$$\rho \approx \frac{4 Q_2}{(Q_2 - \bar{\alpha})^2}. \tag{50}$$

Considering the S-MNIST dataset, Figure 7 shows the bandwidth prior distribution for the semi-supervised (labeled and unlabeled points) and the supervised (labeled points) scenarios.

**Signal Variance $\sigma_f^2$.** Assuming normalized $y_i$, the signal variance $\sigma_f^2$ should be close to 1. Because of this a natural prior for $\sigma^2$ is the truncated normal (onto the set of positive reals $\sigma^2 > 0$), with mode at $1$. The pre-truncation variance can be chosen, for example, to have 0 at 3 standard deviations away from the mode, i.e. it can be chosen to be $1/9$. Note that setting a prior over the signal variance parameter requires evaluating the normalization constant $C_{\nu,\kappa}$[7] for the kernel at each hyperparameter optimization step, something that can be avoided otherwise.

**Noise Variance $\sigma_\varepsilon^2$.** Choosing a prior for the noise variance $\sigma_\varepsilon^2$ is heavily problem-dependent. Truncated normal (onto the set $\sigma_\varepsilon^2 > 0$) with mode at $0$ could be a reasonable option. The pre-truncation variance can be chosen, for example, to be $1$, $1/4$ or $1/9$, placing the value $1$, which is the variance of the normalized observations $y_i$, at 1, 2 or 3 standard deviations away from the mode.

---

[7]Covariance matrix normalization constant $C_{\nu,\kappa}$ can be approximated as $C_{\nu,\kappa} \approx \frac{1}{M} \sum_{i=1}^{M} \boldsymbol{e}_i^T \mathbf{P}_{\mathbf{XX}}^{-1} \boldsymbol{e}_i$, where $M$ is relatively small, $\boldsymbol{e}_i$ are random standard basis vectors and the solve is performed by running the conjugate gradients. Differentiability of the model is preserved. Note however that we did not use this normalization in our tests since the performance improvement we observed was not enough to justify the additional computation overhead. This trade-off can vary considerably from case to case, which is why in our implementation, the covariance normalization is optional.

**Length scale**   The interpretation and the scale of the length scale parameter is manifold-specific. This makes it very difficult to come up with any reasonable prior. Because of this, we suggest actually leaving the length scale parameter free.

**Parameter initialization**   When doing MAP estimation, one can initialize parameters randomly, sampling them from respective priors.

