# OpenReview forum: "Implicit Manifold Gaussian Process Regression"
_NeurIPS.cc/2023/Conference — NeurIPS 2023 poster_

### Official Review · Reviewer_DAU4 · 2023-07-05

**Soundness:** 4 excellent
**Presentation:** 3 good
**Contribution:** 3 good
**Rating:** 7
**Confidence:** 4

**Summary:**

The curse of dimensionality demonstrates in Gaussian process models in that they’re default kernel choices that often depend on the Euclidean distance between 2 points. Euclidean distance is a poor metric especially in high-dimensional settings, where we expect data points to lie on an, often unknown, manifold. This paper combines findings from multiple precursor works in assembling a tool set to for Gaussian process regression with a Metern kernel, while implicitly inferring the underlying manifold structure.

**Strengths:**

I believe that while the paper mostly combines existing theoretical results in manifold learning and numerical analysis, it presents a compelling, novel framework of tackling manifold learning with Gaussian process regression.

The paper is largely self-sufficient: it tackles the complex problem of learning manifolds in the context of Gaussian process by proposing solutions to all facets of the problem: characterizing manifolds with graph Laplacian, gradient-based learning and scalability.

**Weaknesses:**

I think the paper lacks a presentation of _how_ learning the manifold structure aids in the predictive performance of the rotated MNIST dataset. While one can clearly see _why_ the learning of an implicit manifold helps in predicting rotation angles for handwritten digit, as rotation expressly traverses around a nontrivial manifold, the benefits of this work go beyond simple improvements in prediction, but interpretability. While the authors did a very good job in demonstrating how the model correctly inferred the dumbbell-shaped manifold, I believe similar illustrations can also be done in the MNIST rotations.



**Questions:**

None

**Limitations:**

- I believe the presentation of the paper could be improved by a clear separation of predecessor works and original contribution: for example, Section 3.1 is almost solely composed of existing work except for the theoretical contribution of the Matern kernel convergence — an overall marginal contribution in the paper.
- Table 3: “RMSE” in the table seems confusing for prediction of rotating angles. Is it so that the author is referring to the shortest arc as a metric for distance in rotations?
- I believe that the paper could benefit from a illustration of what implicit manifold is learned from the rotated MNIST data: for example, a display of what the kernel values matrix looks like along the trajectory of a rotation compared to the Euclidean kernel matrix, or something else showing the geodesics of the points on the rotation trajectory.

---

> ### Author Rebuttal · Authors · 2023-08-09
>
> *"I believe the presentation of the paper could be improved by a clear separation of predecessor works and original contribution: for example, Section 3.1 is almost solely composed of existing work except for the theoretical contribution of the Matern kernel convergence — an overall marginal contribution in the paper."*
>
> * Thank you for mentioning this point.
> In light of this and some other comments we intend to revisit the presentation in Sections 3 and 4 to make them better structured, emphasize specific algorithm steps and their order, as well as to distinguish our own contributions there from the previous work.
> We believe it will be fairly easy to do this and this will not require major changes.
>
> "*Table 3: “RMSE” in the table seems confusing for prediction of rotating angles. Is it so that the author is referring to the shortest arc as a metric for distance in rotations.*"
>
> * This is a good catch! We actually do use RMSE here although arc length is more appropriate.
> This, however, does not affect the results because the angle of rotation in the dataset was limited to $\pm 45$ degrees.
>
> "*I believe that the paper could benefit from a illustration of what implicit manifold is learned from the rotated MNIST data: for example, a display of what the kernel values matrix looks like along the trajectory of a rotation compared to the Euclidean kernel matrix, or something else showing the geodesics of the points on the rotation trajectory.*"
>
> * Thank you for suggesting the ideas for visualizing the learned manifold.
> Since our method uses the eigenfunctions of the Laplacian as features, it is closely related to the *Laplacian eigenmaps* and the *diffusion maps* dimensionality reduction techniques.
> Using one of these techniques is most likely the best way of visualizing the manifold we are learning.
> Actually, a manifold of rotated images is visualized right in the original paper on diffusion maps: see Figure 2 in Coifman et al. (2006).
> Note though that such a visualization only relies on the first two eigenvectors of the Laplacian---the number of vectors determines the dimension of the "picture" and thus should be no larger than 3---and our kernel uses much more of them.
> We will mention this in the paper.

---

> > ### Comment · Reviewer_DAU4 · 2023-08-10
> > **Post-rebuttal comment**
> >
> > I thank the author for their detailed response, and I maintain the same score assessment but slightly decrease my confidence level as I realize my lack of expertise in manifold learning algorithms such as diffusion maps.
> >
> > I mainly have 1 question regarding the rebuttal PDF: the author added a 3d manifold example in the figure, but I did not see an explanation (it's possible that I missed it in your responses to other reviewers). Could you please clarify what message you meant to convey by this figure?

---

> > > ### Author Response · Authors · 2023-08-11
> > >
> > > Sure. The figures are related to the second point raised by reviewer 2iqe
> > >
> > > "As the methodology applies to general manifolds, I would expect to see more empirical results. For example, some synthetic 3d surfaces..."
> > >
> > > The figures show the results in the semi-supervised learning scenario on a complex 3D surface.
> > >
> > > For the final version of the paper, we chose a 2D example which allows to visualize the model's extension to the ambient space, as depicted in Fig.1 or Fig.3. The visualization of noise presence within manifold samples (Fig. 4) was notably clearer in 2D than in 3D, as well.
> > >
> > > Nevertheless we agree with reviewer 2iqe that a more complicated 3d case might increase further the understanding of the algorithm's features. For instance notice how the posterior standard deviation smoothly decays in proportion to the geodesic distance rather than following the ambient Euclidean distance.
> > >
> > > We plan to add these results in the appendix.

---

### Official Review · Reviewer_g5fB · 2023-07-06

**Soundness:** 4 excellent
**Presentation:** 3 good
**Contribution:** 3 good
**Rating:** 7
**Confidence:** 4

**Summary:**

Authors propose a novel methodology for doing GP regression which is able to learn the implicit structure from the data. This is particularly useful in high-dimensional problems, where the data  lies on low-dimensional manifold. The proposed methods allows to learn the implicit manifold in a fully differentiable way, using nearest neighbour graph.

The method is based on Matern GPs on manifolds and graphs (Borovitskiy, 2020, 2021). To approximate the eigenvalues of the Laplace-Beltrami operator on the implicit data manifold, authors use random walk normalized graph Laplacian (on KNN graph) weighted to overcome possible non-uniform sampling density.

This work leverages efficient approximation to the KNN, sparse structure of the precision matrices, and RFF kernel approximation to be able to scale to large datasets.

The model is tested on a synthetic dumbbell-shape manifold and on rotated MNIST data.

**Strengths:**

This is, to the best of my knowledge, a novel methodology for GP regression. This work is build on solid foundation and the method is derived naturally from Matern GPs on manifolds and graphs. Authors cleverly use a variety of techniques to keep good scalability.

In my opinion, the main contribution of this work is in the elegant assembly of the right methods, which gives a practically usable model for a well-known problem.

The manuscript is written clearly and is easy to follow.

**Weaknesses:**

Due to the incremental nature of the paper, it is not always clear what is a novel contribution of this work and what is a background from previous works (example, last paragraph of 2.1).

The experiments are somewhat limited. The method is tested on one synthetic dataset and a few versions of rotated MNIST. In the MNIST dataset, however, the added rotation must be inducing a lot of structure. It would be beneficial to showcase the method on a real dateset where an implicit manifold is already existing in the data.

**Questions:**

1. Could you please clarify on the following: There are no results on spectral convergence for KNN graphs independent of data sampling density. You use a weighing scheme to deal with non-uniform data sampling. You do not claim that it implies spectral convergence. How much of a problem is it, both theoretically and practically?

Suggestions:
1. In figure 2, perhaps use different colours foe kernel and sample plots.
2. eq(11) I believe $\lambda_i$ should be $\lambda_l$?

**Limitations:**

The limitations are adequately addressed.

---

> ### Author Rebuttal · Authors · 2023-08-09
>
> *"it is not always clear what is a novel contribution of this work and what is a background from previous works (example, last paragraph of 2.1)."*
>
> * Thank you for mentioning this point.
> In light of this and some other comments we intend to revisit the presentation in Sections 3 and 4 to make them better structured, emphasize specific algorithm steps and their order, as well as to distinguish our own contributions there from the previous work.
> We believe it will be fairly easy to do this and this will not require major changes.
>
> "*The experiments are somewhat limited. The method is tested on one synthetic dataset and a few versions of rotated MNIST. In the MNIST dataset, however, the added rotation must be inducing a lot of structure. It would be beneficial to showcase the method on a real dateset where an implicit manifold is already existing in the data.*"
>
> * In addressing the high-dimensional context, we employed the rotated MNIST dataset where we have good reasons to believe that the manifold hypothesis holds. Other high-dimensional datasets may lack an explicit enough underlying manifold structure. The attached PDF (see general response) contains a table presenting preliminary results in the supervised scenario for a random dataset from the UCI Machine Learning Repository. RMSE performance with IMGP surpassed that of EGP for larger sample sizes. However, NLL exhibited a less favorable trend compared to our observations for rotated MNIST. This warrants further investigation. It is possible that the chosen dataset, despite the high-dimensional feature space, does not inherently embody a manifold structure or such a structure could be highly irregular. This is supported by the need for a substantial number of eigenpairs in achieving satisfactory outcomes in contrast to rotated MNIST i.e. slow spectrum decay. We will discuss this example or a similar additional example in the paper as well as an example for a manifold in 3D---the results for the latter were actually excluded from the draft to save space. We think it might make sense to apply the technique in the contexts like modeling 3d structures of molecules where there is inherent symmetry (invariance or equivariance to rotations and translations), as an interesting direction for further work.
>
> *"Could you please clarify on the following: There are no results on spectral convergence for KNN graphs independent of data sampling density. You use a weighing scheme to deal with non-uniform data sampling. You do not claim that it implies spectral convergence. How much of a problem is it, both theoretically and practically?"*
>
> * We hypothesize that convergence should still hold true.
> There is apparently no proof of this fact in the literature though.
> It does not seem to be out of reach for the proof techniques used for similar problems, but such a proof would likely turn out to be long and heavy due to multiple "moving parts".
>
> * Practically, we do not expect this to be a problem either.
> Although we use convergence results to motivate the technique, the graph construction itself reflects the geometry of a point cloud in a rather intuitive way.
> What is more, it easy to imagine that the graph is capable of representing structures beyond manifolds in the rigorous mathematical sense, e.g. manifolds of different dimensions glued together (like presented on Figures 2--4 in Dunson et al. (2022).
>
> *"In figure 2, perhaps use different colours foe kernel and sample plots."*
>
> * Thank you, we agree this should improve clarity.
> We will make this change for the camera ready version.
>
> *"eq(11) I believe  should be ?"*
>
> * This is an actual typo.
> Thank you very much for reporting this!

---

### Official Review · Reviewer_2ieq · 2023-07-07

**Soundness:** 2 fair
**Presentation:** 3 good
**Contribution:** 3 good
**Rating:** 5
**Confidence:** 4

**Summary:**

The authors propose a methodology to extend Matern processes on implicit manifolds, which are modeled by $K$-NN graphs, and the kernel relies on the set of eigenvalues/vectors of the associated graph-Laplacian. An approximation of the eigenfunctions based on the eigenvectors is provided, which together with an additional trick allows to extend the process in the vicinity of the implicit manifold. The hyperparameters are learned in supervised and semi-supervised scenarios via a differentiable objective, and the performance is demonstrated in the experiments.

**Strengths:**

- The technical part of the paper seems correct and sensible, but I have not checked the derivations in details.
- The idea is a direct and relatively simple extension of previous work.
- The results mainly due to the synthetic experiment seem convincing.
- The paper in general is well-written.

**Weaknesses:**

- The main drawback of graph-based methods is typically the construction of the graph and how well this recovers the actual structure of the underlying manifold. I suppose that the current method has the same limitation that is not clearly discussed/analyzed in the paper.
- As the methodology applies to general manifolds, I would expect to see more empirical results. For example, some synthetic 3d surfaces and more complicated high-dimensional real-world datasets as the rotation of MNIST is closer to a synthetic experiment.


**Questions:**

Q1. Lines 98-100: Is there any reference that plugging in Eq. 3 the graph Laplacian and the Gaussian noise gives a Matern process on the graph?

Q2. Why the dimension of the implicit manifold (graph) is 0? I suppose that the underlying manifold has some dimensionality. I think that there are approaches relying on graphs that try to estimate this dimensionality.

Q3. How sensitive is the method with respect to the graph construction? In particular, how the number of neighbors $K$ influences the result? Can this be estimated somehow e.g. by cross-validation?

Q4. How the method behaves if there are more than one connected components in the dataset?

Q5. Lines 144-150: The graph Laplacian converges to the Laplace-Beltrami (in terms of eigenvalues/vectors/functions) for $K$-NN graphs as long as the data distribution is uniform on the manifold? While for non-uniform distributions such a convergence result does not exist?

Q6. Lines 177-179: I think that here the gist is unclear.

Q7. Which Nyström method is used?

Q8. Fig 3. I think it is interesting to see the actual eigenvector on the data (e.g. color the points) and how this correlates to the extended version.

Q9. Eq. 11 the eigenvalue should be probably $\lambda_l$ and not $\lambda_j$?

**Limitations:**

The authors do not discuss the limitations of their work, and in particular, the construction of the graph which I think is critical for the performance of the model. The methodology that is propose does not have a direct negative societal impact.

---

> ### Author Rebuttal · Authors · 2023-08-09
>
> *"the construction of the graph and how well this recovers the actual structure of the underlying manifold"*
>
> * This is indeed a very appropriate valid concern.
> We apologize for not explicitly discussing the limitations in the paper.
> We sketched a paragraph on this (see general response) and aim to include it into the new version of the paper.
> KNN graph can fail to capture the geometry.
> Because of the asymptotic convergence, it should not be a problem when there is a lot of (potentially unlabeled) data.
> This is why we emphasize the semi-supervised case where large number of unlabeled data points can facilitate small data learning from the labeled ones.
> Furthermore, our framework, being fully differentiable, sets up the groundwork for more sophisticated techniques that might include *graph learning* as in, for example, Kazi et al (2022).
> In this particular paper, however, we wanted to describe the simple model and lay groundwork for the possible extensions.
>
> *"The paper would greatly benefit from additional empirical findings."*
>
> * In addressing the high-dimensional context, we employed the rotated MNIST dataset where we have good reasons to believe that the manifold hypothesis holds.
> Other high-dimensional datasets may lack an explicit enough underlying manifold structure.
> The attached PDF (see general response) contains a table presenting preliminary results in the supervised scenario for a random dataset from the UCI ML Repository.
> RMSE performance with IMGP surpassed that of EGP for larger sample sizes.
> However, NLL exhibited a less favorable trend compared to our observations for rotated MNIST.
> This warrants further investigation.
> It is possible that the chosen dataset, despite the high-dimensional feature space, does not inherently embody a manifold structure or such a structure could be highly irregular.
> This is supported by the need for a substantial number of eigenpairs in achieving satisfactory outcomes in contrast to rotated MNIST i.e. slow spectrum decay.
> We will discuss this example or a similar additional example in the paper as well as an example for a manifold in 3D (results in this case are also present in the attached pdf) that we excluded before to save space.
>
> *"Q1"*
>
> * Yes, this is discussed in detail in Borovitskiy et al. (2021).
> This reference was accidentally removed from lines 98-100 in one of the final versions of the draft.
> Thank you very much for spotting this!
>
> *"Q2"*
>
> * There is plenty of ways to estimate $\operatorname{dim}(M)$, from classical(Levina and Bickel, 2004) to modern (Denti et al. 2022).
> We do not estimate $\operatorname{dim}(M)$ though, for multiple reasons.
> First and foremost, because this merely results in  a reparameterization of the Matérn family: the meaning of parameter $\nu$ becomes different, but in any case it is an unknown parameter to be chosen somehow.
> Second, the possible values of $\nu$ will be usually restricted by the problem size and computational resources at hand because larger $\nu$ implies higher costs.
> A reasonable approach is to choose $\nu$ by grid search where the grid consists of small integer values $1, 2, 3, \ldots$.
>
> *"Q3"*
>
> * Larger values of $K$ should in principle---and usually in practice, as we observed---lead to better results: the weighting of the graph does the heavy-lifting of capturing the geometry.
> As a rule of thumb you should use $K$ as large as you can afford.
>
> *"Q4"*
>
> * The KNN graph we are using is always connected.
> We further use such bandwidth prior that prevents weights from being overly small.
> To consider datasets with well separated connected components, one can consider clustering them and then applying our technique to each of the clusters separately.
> This would correspond to the geometric assumption of having a disconnected manifold.
> This would, however, often make little sense because the aforementioned implies zero correlation between points on different connected components, i.e. this would correspond to modeling the connected components independently.
> Some datasets may posses a hierarchical structure: have many (small) disconnected components which, as whole components, covary with one another.
> This might require some modified geometric assumptions (and model) to handle.
>
> *"Q5"*
>
> * We expect that convergence still holds for KNN graphs even in the case of a non-uniform density (when using Coifman's normalized adjacency matrix $\mathbf{A} = \tilde{\mathbf{D}}^{-1} \tilde{\mathbf{A}} \tilde{\mathbf{D}}^{-1}$), but such a result cannot be found in the literature, to the best of our knowledge.
> When the distribution is uniform and using the usual adjacency matrix ($\mathbf{A} = \tilde{\mathbf{A}}$) the convergence for KNN graphs is studied, for example, in Calder and Trillos (2022).
>
> *"Q6"*
>
> * Thank you for spotting this.
> We realize now that this statement is unclear, the term "weighting scheme" is nor very intuitive nor is it introduced in any way.
> What is meant here is that, following Coifman et al. (2006), we (1) use the random-walk normalized Laplacian, as opposed to the unnormalized one or the symmetric normalized one and (2) use the normalized adjacency matrix $\mathbf{A} = \tilde{\mathbf{D}}^{-1} \tilde{\mathbf{A}} \tilde{\mathbf{D}}^{-1}$ to counteract the presence of non-uniform density---using this adjacency is what we meant by the "weighting scheme" because it defines graph weights.
>
> *"Q7"*
>
> * Like, for example, in Section 3 of https://proceedings.neurips.cc/paper_files/paper/2003/file/cf05968255451bdefe3c5bc64d550517-Paper.pdf.
>
> *"Q8"*
>
> * Thank you for this suggestion. We did not include such a picture because of continuity: the values of the eigenvector match the values of the extension close to the manifold. We are happy to add the picture you requested to the appendix.
>
> *"Q9"*
>
> * This is a typo indeed, thank you very much for spotting this!
>
> *"The authors do not discuss the limitations of their work"*
>
> * Thank you for mentioning this. Please see the general response.

---

> > ### Comment · Reviewer_2ieq · 2023-08-16
> > **Post-rebuttal comment**
> >
> > I would like to thank the authors for their responses and the additional demonstrations. Regarding the graph construction, I agree that having more data implies that $K$ becomes a less critical parameter, but instead, I believe that the bandwidth of the kernel becomes critical for capturing well the structure of the underlying manifold with the graph. Anyways, this is a classic issue in graph construction, and the current papers focus on a different problem. I also agree that the manifold assumption does not hold in many of the real-world datasets. I recommend the authors include in the updated paper 3D demonstrations, potentially showing the eigenvectors together with the extended version (e.g. in Fig.3 ), and also discussing clearly the limitations of the proposed approach.
> >
> > As other reviewers mentioned, the novelty of the paper is somewhat limited in the sense that it combines previous approaches in a practical way. Overall and in light of the other reviews with there being a consensus for acceptance, I increase my score and vote for borderline acceptance.

---

> > > ### Author Response · Authors · 2023-08-16
> > >
> > > We thank the referee for raising the score and supporting paper acceptance. We will implement the suggested changes in the camera ready version.

---

### Official Review · Reviewer_ejKa · 2023-07-13

**Soundness:** 3 good
**Presentation:** 3 good
**Contribution:** 3 good
**Rating:** 6
**Confidence:** 4

**Summary:**

This work extends the reach of Matern Gaussian processes to additionally learning the implicit low-dimensional (unknown) manifold the data lives on - the existence of such a manifold is suggested by the manifold hypothesis. The theory draws from existing Laplacian Matern Gaussian processes (Borovitskiy et al., 2020). The model they propose is differentiable wrt all model and geometry hypers. It is able to scale to thousands of data points by leveraging the sparse structure of Mather precision matrices. They provide a way to extend predictions to the whole R^{d} space by reverting to a Euclidean GP away from the manifold.  The experimental evaluation shows support for the implicit manifold Gaussian process.

**Strengths:**

- The end-to-end differentiability where the kernel hyperparameters along with those that parameterise the underlying geometry can be learnt simultaneously using a single objective.
- Scalability is achieved as the precision matrix corresponding to the KNN implied graph is sparse.
- The semi-supervsied case is interesting where a large amount of unlabelled data is leveraged to infer the underlying geometry of the manifold through a weighted nearest neighbour  graph.
All these points contribute to the quality and significance of the work.

**Weaknesses:**

- The exposition could benefit from an algorithm style pseudocode for the training and prediction steps, it should start with the interpretation of the data as a graph, computation of the elements needed for the Matern kernel etc - as I am a bit confused about the order of the steps.

- While the details are all there in section 3.2, I think there need to be separate sections for construction of the matern kernel on graph nodes, computing the dependencies ie. the eigenpairs of the graph Laplacian, computing the kernel on arbitrary vectors in R^{d}, computing the predictive posterior.

- I dont understand this line - pls explain or rewrite. Basically, why dont you just say how do you compute dim(M)?

Essentially, from the theoretical results of Section 3.1 we borrow a particular weighting scheme, aiming to cancel out the possibly non-uniform density, and a specific choice of the graph Laplacian (the random walk normalized one).

**Questions:**

- The final predictive distribution is a mixture of Gaussians - one is the canonical Euclidean posterior predictive with SE-ARD kernel and the other is the geometry aware kernel underlying the same canonical posterior predictive equations?

- What is the order of the steps in training - the log marginal likelihood depends entails the precision matrix which depends on the eigenpairs, but you say in line 206 that they are computed after the hyperparameters are found, these are mentioned in line 216 - is the \hat{\kappa} the number of neighbours of the KNN graph?

**Limitations:**

I think authors should devote a paragraph to this. I don't see limitations discussed anywhere.

---

> ### Author Rebuttal · Authors · 2023-08-09
>
> *"The exposition could benefit from an algorithm style pseudocode ..", ".. I am a bit confused about the order of the steps.", "What is the order of the steps in training"*
>
> * Thank you for mentioning this.
> We will emphasize the general flow of the algorithm.
> To clarify, the algorithm can be summarized as follows.
> 1. We compute KNN index using FAISS. This is enough to define matrix-vector products (MVPs) with the weighted adjacency matrix of the graph.
> 2. We find the optimal hyperparameters $\hat{\mathbf{\theta}}$ by maximizing the likelihood (14). Here we assume that $\nu$ is a small integer and use Proposition 2 to efficiently compute MVPs with the precision matrix using MVPs with the adjacency.
> 3. We compute a set of eigenpairs corresponding to the smallest eigenvalues of the Laplaican (with fixed hyperparameters $\hat{\mathbf{\theta}}$) using the Lanczos algorithm.
> 4. We define the kernel as
>     $$
>     k(\mathbf{x}, \mathbf{x}') = \frac{\hat{\sigma_f}^2}{C_{\hat{\nu}, \hat{\kappa}}} \sum_{l=1}^L \Phi_{\hat{\nu}, \hat{\kappa}}(\lambda_l) f_l(\mathbf{x}) f_l(\mathbf{x}'), \qquad \Phi_{\nu, \kappa}(\lambda) = \left(\frac{2 \nu}{\kappa^2} + \lambda\right)^{-\nu}
>     $$
>     where $f_l$ are the Laplacian eigenvectors extended to the whole $\mathbb{R}^d$ via (11).
> 5. We compute the posterior $f^{(m)}$ corresponding to the kernel $k$. We also perform Gaussian process regression (including hyperparameter tuning) with the Euclidean squared exponential kernel, we call the result $f^{(e)}$. The final predictive model is then given by their weighted average (12).
>
>
> *"While the details are all there in section 3.2, I think there need to be separate sections for construction of the matern kernel on graph nodes, computing the dependencies ie. the eigenpairs of the graph Laplacian, computing the kernel on arbitrary vectors in $R^{d}$, computing the predictive posterior."*
>
> * Thank you for suggesting how we can make the presentation more clear.
> This suggestion is clearly connected to the one above and to the questions some other referees have.
> We will revisit the presentation in Sections 3 and 4 to make them better structured, emphasize specific algorithm steps and their order, as well as to distinguish our own contributions there from the previous work.
> We believe it will be fairly easy to do this and this will not require major changes.
>
> *"I dont understand this line - pls explain or rewrite."*
>
> * Thank you for spotting this.
> We realize now that this statement is unclear, the term "weighting scheme" is nor very intuitive nor is it introduced in any way.
> What is meant here is that, following Coifman et al. (2006), we (1) use the random-walk normalized Laplacian, as opposed to the unnormalized one or the symmetric normalized one and (2) use the normalized adjacency matrix $\mathbf{A} = \tilde{\mathbf{D}}^{-1} \tilde{\mathbf{A}} \tilde{\mathbf{D}}^{-1}$ to counteract the presence of non-uniform density---using this adjacency is what we meant by the "weighting scheme" because it defines graph weights.
>
> *"how do you compute dim(M)?"*
>
> * There is plenty of ways to estimate $\operatorname{dim}(M)$, from classical (Levina and Bickel, 2004) to modern (Denti et al. 2022).
> We do not estimate $\operatorname{dim}(M)$ though, for multiple reasons.
> First and foremost, because this merely results in  a reparameterization of the Matérn family: the meaning of parameter $\nu$ becomes different, but in any case it is an unknown parameter to be chosen somehow.
> Second, the possible values of $\nu$ will be usually restricted by the problem size and computational resources at hand because larger $\nu$ implies higher costs.
> A reasonable approach is to choose $\nu$ by grid search where the grid consists of small integer values $1, 2, 3, \ldots$.
>
> *The final predictive distribution is a mixture of Gaussians - one is the canonical Euclidean posterior predictive with SE-ARD kernel and the other is the geometry aware kernel underlying the same canonical posterior predictive equations?*
>
> * Almost, with one clarification, the final predictive distribution is a *Gaussian distribution*, not a mixture of Gaussians.
> Its mean and variance are convex combinations of the mean and variance of the canonical Euclidean posterior and the posterior under the geometry aware kernel similar to what you suggested.
> Another point to note is that in our experiments we did not use the ARD kernel, just the plain kernel with one length scale although nothing prevents you from using it.
>
> *"the log marginal likelihood depends entails the precision matrix which depends on the eigenpairs, but you say in line 206 that they are computed after the hyperparameters are found, these are mentioned in line 216"*
>
> * You are right that the precision matrix can be (approximately!) expressed in terms of the eigenpairs corresponding to the smallest eigenvalues of the Laplacian.
> However, thanks to Proposition 2, for integer values of $\nu$ the precision matrix can also be represented (exactly!) as a polynomial of the Laplacian. This allows us to efficiently compute exact matrix-vector products with the precision without evaluating the eigenpairs on each optimization step, and to find the hyperparameters without inefficient differentiation through eigenpair-computing routines.
>
> *"is the $\hat{\kappa}$ the number of neighbors of the KNN graph?"*
>
> * No, $\hat{\kappa}$ is the length scale of the geometric kernel, after optimization.
>
> *"I think authors should devote a paragraph to this. I don't see limitations discussed anywhere."*
>
> * Thank you for mentioning this. Please see the general response.

---

> > ### Comment · Reviewer_ejKa · 2023-08-20
> > **Post rebuttal comment**
> >
> > I would like thank the authors for their response and clarifications. I am happy to support this work and raise my confidence score to a 4 and keep my overall score intact at 6.
> >
> > I would urge the authors to restructure section 3.2, clarify the training algorithm and ensure that all the parameters are described before being used in equations, for example \hat{\kappa}.

---

### Author Rebuttal · Authors · 2023-08-09

We thank the referees for their valuable summaries and insights which will help us improve the paper.
We have replied to every review with detailed comments.

An essential suggestion made by many referees was to add an explicit discussion of limitations.
One major limitation is that there is a trade-off between the complexity of geometry and non-uniformity of sampling data on one side and the number of unlabeled data points needed to capture meaningful structure on the other side.
The model works well when the former is not too pronounced and the latter is large.
Further, the model could struggle for clustered data.
Possible avenues for improving on the mentioned drawbacks include learning a graph in a more sophisticated way (e.g. adaptive bandwidth, differentiable graph learning) or clustering and combining models.
Another limitation is that using larger values of $\nu$ and $K$ result that may be desirable sometimes imply higher costs and are not always feasible.
We fully agree that including this explicitly would improve paper quality and aim to include it in the camera-ready version.

---

### Decision · Program_Chairs · 2023-09-21

**Decision:**

Accept (poster)

**Comment:**

Four knowledgeable referees support accept, and I accept.